# Increased AID results in mutations at the *CRLF2* locus implicated in Latin American ALL health disparities

Valeria Rangel[1,2,12], Jason N. Sterrenberg[1,12], Aya Garawi[3], Vyanka Mezcord[4], Melissa L. Folkerts [1,2], Sabrina E. Calderon[3], Yadhira E. Garcia[5], Jinglong Wang[6], Eli M. Soyfer [1,2], Oliver S. Eng [7,8], Jennifer B. Valerin[1,8], Sora Park Tanjasiri[8,9], Fabiola Quintero-Rivera[10,11], Marcus M. Seldin [2,8], Selma Masri [2,8], Richard L. Frock [6], Angela G. Fleischman [1,2,8] & Nicholas R. Pannunzio [1,2,8] ✉

Activation-induced cytidine deaminase (AID) is a B cell-specific mutator required for antibody diversification. However, it is also implicated in the etiology of several B cell malignancies. Evaluating the AID-induced mutation load in patients at-risk for certain blood cancers is critical in assessing disease severity and treatment options. We have developed a digital PCR (dPCR) assay that allows us to quantify mutations resulting from AID modification or DNA double-strand break (DSB) formation and repair at sites known to be prone to DSBs. Implementation of this assay shows that increased AID levels in immature B cells increase genome instability at loci linked to chromosomal translocation formation. This includes the *CRLF2* locus that is often involved in translocations associated with a subtype of acute lymphoblastic leukemia (ALL) that disproportionately affects Hispanics, particularly those with Latin American ancestry. Using dPCR, we characterize the *CRLF2* locus in B cell-derived genomic DNA from both Hispanic ALL patients and healthy Hispanic donors and found increased mutations in both, suggesting that vulnerability to DNA damage at *CRLF2* may be driving this health disparity. Our ability to detect and quantify these mutations will potentiate future risk identification, early detection of cancers, and reduction of associated cancer health disparities.

Diversification of the immunoglobulin heavy chain (*IGH*) locus in B cells is a form of physiological genome editing that has been conserved over 500 million years of evolution[1,2]. As B cells mature, DNA double-strand breaks (DSBs) are generated at two different points by two distinct mechanisms to modify the region of the *IGH* gene that encodes for either the antibody variable region or the antibody constant region[3]. In pro-B/pre-B cells, the recombination activating gene (RAG) complex is expressed to generate DSBs at the variable (V), diversity

[1]Division of Hematology/Oncology, Department of Medicine, University of California, Irvine, Irvine, CA, USA. [2]Department of Biological Chemistry, University of California, Irvine, Irvine, CA, USA. [3]School of Biological Sciences, University of California, Irvine, Irvine, CA, USA. [4]Center for Applied Biotechnology Studies, Department of Biological Science, California State University Fullerton, Fullerton, CA, USA. [5]Department of Pharmaceutical Sciences, School of Pharmacy & Pharmaceutical Sciences, University of California, Irvine, CA, USA. [6]Division of Radiation and Cancer Biology, Department of Radiation Oncology, Stanford University School of Medicine, Stanford, CA, USA. [7]Division of Surgical Oncology, Department of Surgery, University of California, Irvine, Irvine, CA, USA. [8]Chao Family Comprehensive Cancer Center, University of California, Irvine, Irvine, CA, USA. [9]Department of Health, Society and Behavior, University of California, Irvine, Irvine, CA, USA. [10]Department of Pathology and Laboratory Medicine, University of California, Irvine, Irvine, CA, USA. [11]Department of Pediatrics, University of California, Irvine, Irvine, CA, USA. [12]These authors contributed equally: Valeria Rangel, Jason N. Sterrenberg. ✉e-mail: nrpann@hs.uci.edu

(D), and joining (J) gene segments during V(D)J recombination[4]. In mature B cells, activation-induced cytidine deaminase (AID) is expressed to initiate the processes of class-switch recombination (CSR) and somatic hypermutation (SHM)[5,6].

AID evolved to be an integral part of the adaptive immune system[7], but its expression must be tightly sequestered to mature B cells as off-target deamination events have the potential to result in mutations genome-wide. The consequences of aberrant AID expression and off-targeting are evident in the etiology of several B cell cancers[3,8–10]. Further, that AID is B cell specific, and B cell cancers account for approximately 80-90% of hematopoietic malignancies[3,11], is a further indicator of the dangers of abnormal AID activity. Sequestering the RAG-induced DSBs of V(D)J recombination to pre-B cells and the AID-induced DSBs of CSR to mature B cells helps maintain genome stability. However, AID can be transiently expressed in pre-B cells concurrent with RAG expression when pre-B cells are stimulated by cytokines[12]. Indeed, examination of the various chromosomal translocations that occur in many lymphoid malignancies show that one DSB was RAG-induced at the *IGH* locus adjacent to a D or J cassette while the other DSB locus shows evidence of AID activity[13–15], indicating that the most likely mechanism for oncogenesis in a subset of blood cancers is AID expression in pre-B cells[10,12,16].

Understanding both why some RAG-induced DSBs in pre-B cells fail to be efficiently repaired by non-homologous end-joining (NHEJ) and what leads to AID activity in pre-B cells will allow us to better predict populations susceptible to certain blood cancers and discern the etiology of these malignancies. Two loci that show mutations consistent with AID activity in pre-B cells are *BCL2* and *CRLF2*[13–15]. Unlike random DSB sites that can span tens or hundreds of kilobases, these AID-induced DSBs are focal with many occurring within 20-600 bp regions[10] that we refer to as AID break clusters (ABCs).

A critically understudied area in genome instability associated with diseases are cancer health disparities where one racial or ethnic group shows higher incidence and poorer outcomes of certain cancer types[17,18]. For example, Philadelphia chromosome-like acute lymphoblastic leukemia (Ph-Like ALL) has a higher cancer incidence in people that self-identify as Hispanic and have genetic similarities with Latin Americans[19–21]. Ph-like ALL shares several characteristics with Philadelphia-positive (Ph[+]) ALL regarding its gene expression profile, yet lacks the eponymous t(9;22) *BCR::ABL1* translocation[22]. Hispanics and Latinos specifically have a higher likelihood of developing Ph-like ALL and respond less favorably to current standard treatments[19,23]. Genome rearrangements and mutations leading to Ph-like ALL are highly diverse, yet Hispanics and Latinos are more likely to have Ph-like ALL characterized by a translocation involving *IGH* and the ABC upstream of the *CRLF2* locus[19,23]. This is in contrast to Europeans and non-Latin Americans that more often have an interstitial deletion between *CRLF2* and *P2RY8* leading to a gene fusion[24]. While some risk alleles have been identified that link Hispanics to higher Ph-like ALL risk[12], no molecular mechanism has been established to understand why *CRLF2::IGH* formation is more prevalent in this population.

Here, using both human cell lines and patient samples we identify an increase in insertion and deletion events (indels) at two ABCs, the *CRLF2* ABC linked to Hispanics with Ph-like ALL and the *BCL2* ABC associated with follicular lymphoma[17]. In many cases, these indels are the "mutation scars" left behind by NHEJ DSB repair due to processing factors that remove or add nucleotides prior to ligation[17,25]. To deeply characterize these genetic changes, we developed a digital PCR (dPCR) assay to detect and quantify mutational scarring at sites targeted by AID. We show that increased AID expression correlates with an increase in mutations at these ABCs. Strikingly, these mutations are also detected in ALL patient samples and at the *CRLF2* ABC in healthy Hispanic donors, indicating that off-target activity of AID at *CRLF2* occurs during B cell maturation and can be a risk factor of Ph-like ALL development.

## Results

### Amplicon sequencing (AmpSeq) of ABCs shows increased indels in human B cells

Of the known ABCs associated with B cell malignancies[3], the major focus here is on the ABC upstream of the *CRLF2* gene[14]. *CRLF2* is located on a pseudoautosomal region (PAR) shared by both the X and Y chromosomes[26]. We also include data on the ABC in the 3' UTR of the *BCL2* gene that is also called the *BCL2* Major Breakpoint Region (MBR) (Fig. 1A) to demonstrate that more than one ABC site can be targeted even though only a DSB at one leads to an oncogenic translocation with *IGH*[13].

According to the mapping of chromosomal translocation junctions in human patients, a DSB that results in pathogenic rearrangements involving *CRLF2* can occur anywhere in the 27 kb region upstream of the gene, yet two specific DSB clusters are of note[14]. Approximately 4.5 kb upstream of *CRLF2* is a cluster that shows the CAC/CACA motif associated with being a cryptic recombination signal sequence (RSS) recognized by the RAG complex[14,27,28]. Off-target RAG cutting at this site leads to an intra-chromosomal deletion that puts the *CRLF2* gene under control of the *P2RY8* promoter, leading to its overexpression[14,29]. The other site is 16 kb upstream of *CRLF2* and contains no cryptic RSS sites, but does have DSBs at or near a CpG, which is a hallmark of ABCs since AID has an affinity to deaminate cytosines within RCG sequences (WGCW» > WRC»RCG)[30–32]. For rearrangements involving ABCs, it has been shown that a translocation is 30% more likely to occur directly at a CpG dinucleotide and 70% more likely to occur within 8 bp of a CpG dinucleotide[13].

AID requires single-stranded DNA (ssDNA) as a substrate for deamination[30]. AID expression in the human pre-B cell lines Nalm6 and Reh is undetectable[33], but ssDNA generated at these sites would still be vulnerable to damage. Thus, we tested if ABCs show an inherent propensity for damage even in the absence of AID. This was done using amplicon sequencing (AmpSeq). The detailed sequence for the *CRLF2* region with the ABC (ChrX: 1,228,310-1,228,969, hg38) is shown in Supplementary Fig. 1A. Strikingly, as with the reported patient data[13,14], there is a peak of deletions near the CpGs in the *CRLF2* ABC in both Nalm6 and Reh cell lines (Fig. 1B). These deletions likely represent mutation scars from DSB formation and NHEJ repair, which is prone to creating indels. In Nalm6 cells, indels appear very focal and are highest at the first two CpG sites. While indels are also highly clustered at this site in Reh cells, we found that the indels appear more spread throughout the ABC. In both cell lines the downstream CpG does not appear to accumulate indels.

Similarly, using AmpSeq at the *BCL2* ABC (Chr18: 63,126,040-63,126,870, hg38) (Supplementary Fig. 1B), we also see indels, yet observe a notably different pattern for each cell line (Fig. 1C). It has been reported that DSBs cluster at the three regions carrying CpG sites in patients[13]. In Nalm6 cells, we measure a major peak centered on the middle CpG site, but no indels at the flanking CpG sites. In contrast, while there is still a peak in Reh cells, it is downstream of the three CpG sites. This may be indicative of either this region being less accessible to enzymes that drive formation of these indels or that DSB ends are processed or resected differently once formed. Importantly, that indels accumulate in cell lines near CpGs that correlate with DSBs involved in chromosomal translocations in human patients[13,14] shows these regions, even without AID, have an innate instability that has been linked to their cytosine content[34].

### CRISPR/Cas9 demonstrates ability of dPCR to detect indels at ABCs

While AmpSeq can provide an analysis of sequence changes occurring at ABCs, we sought an improved method that was more rapid and quantifiable to detect mutations within ABCs and developed an innovative dPCR-based assay[35] for the *CRLF2* ABC. Genomic DNA (gDNA) is isolated from human pre-B cell lines or whole blood from patients and

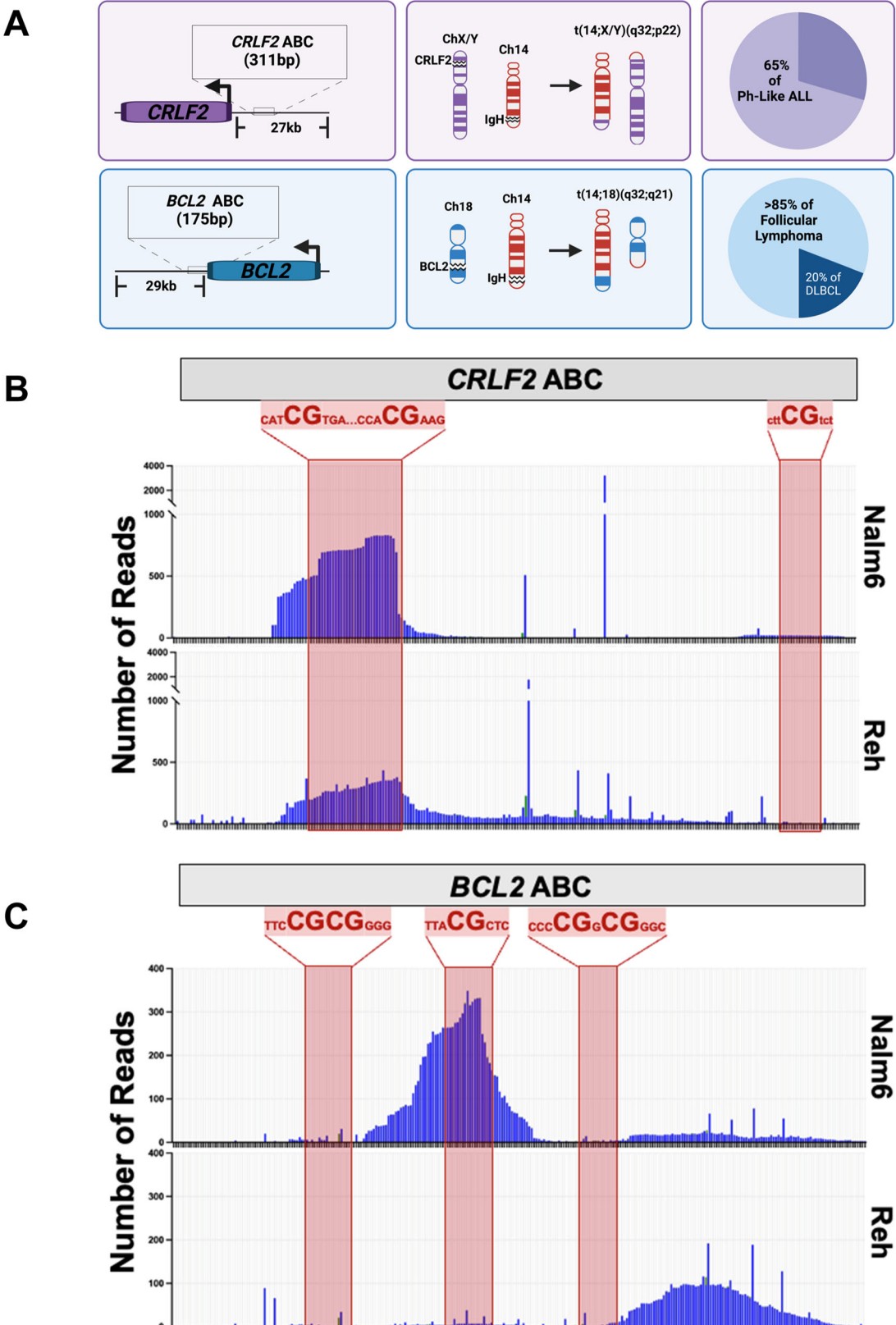

subjected to dPCR where absolute quantification of fluorescent Taq-Man probe binding to a generated amplicon provides details of mutations occurring at ABC sites (Fig. 2A). For *CRLF2*, three TaqMan probes bind directly at one of the three AID-preferred CpG sites (FAM, TAMRA, Cy5); the SUN probe binds where no CpG sites or DSBs have been mapped and acts as a detector for mutations at a non-AID target

(Fig. 2B, Supplementary Fig. 1A). These are referred to as drop-off probes since mutation or DSB formation and repair would alter the sequence, preventing probe binding (i.e., it "drops-off").

First, we wanted to test the sensitivity of the dPCR assay by inducing DSBs at the ABC sites. Single-guide RNAs (sgRNAs) were designed to direct Cas9 to *CRLF2* on chromosome X/Y (Supplementary

**Fig. 1 | AID Break Clusters (ABCs) associated with recurrent translocations in B cell cancers show evidence of genome instability. A** Diagram of *CRLF2* and *BCL2* ABC sites and their locations relative to their corresponding gene. The oncogenic translocations formed between each of the two ABC sites and *IGH* are represented in the middle column, where pathological (AID) and physiological (RAG) double-stranded breaks are portrayed by open gaps in the chromosomes. The percentage of the B cell cancer subtypes associated with the oncogenic translocations are shown in the last column with data taken from[3,23,49]. Insertion and deletion (indel) events detected by AmpSeq at **B** *CRLF2* and **C** *BCL2* ABC sites in Nalm6 (top panel) and Reh (bottom panel) cells. CpG sites spanning each of the three ABC sites are

represented by the partial nucleotide sequence shown above the red highlighted regions overlaying the amplicon sequencing data (see Supplementary Fig. 1 for detailed sequence information). Each bar represents changes at a single base pair with deletions (blue) more dominant than insertions (green). Based on the December 2013 GRch38/gh38 build of the human genome, the coordinates for the *CRLF2* ABC are ChrX:1,228,310-1,228,969 (same for the Y chromosome) and the coordinates for the *BCL2* MBR ABC are Chr18:63,126,040-63,126,870. Figure 1A created with BioRender.com released under a Creative Commons Attribution-NonCommercial-NoDerivs 4.0 International license. Source data are provided as a Source Data file.

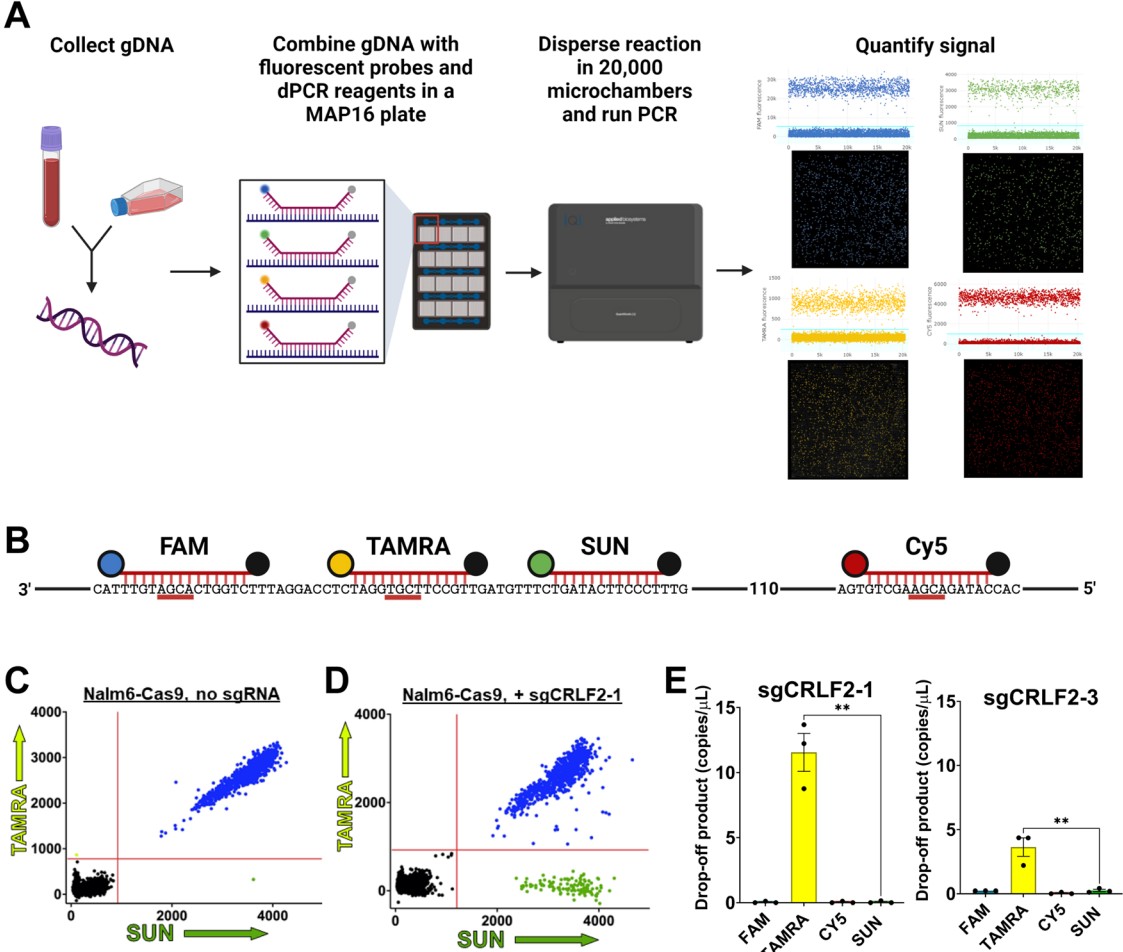

**Fig. 2 | Digital PCR (dPCR) can detect and quantify indels following DSB formation and repair at the *CRLF2* ABC. A** Workflow for dPCR assay using the Applied Biosystems Absolute-Q. Prepared gDNA is combined with amplicon primers, TaqMan probes, and master mix then loaded onto a MAP16 plate where the reaction is partitioned into over 20,000 microchambers. A representative experiment showing quantification of fluorescent signal from 4 channels indicates the detected signal from each microchamber. **B** Schematic of FAM, TAMRA, Cy5 and SUN drop-off probe binding sites within the *CRLF2* ABC region. **C** dPCR results in cells with no Cas9 cutting. Results are shown as a 2D scatterplot with the fluorescent signal from each of 20,000 individual microchambers indicated. Colors represent SUN alone (green) TAMRA alone (yellow/green), SUN + TAMRA (blue), or microchambers with no signal (black). While only TAMRA and SUN are shown here, fluorescent signals from all four probes are gathered simultaneously. **D** dPCR

results as a 2D scatter plot following Cas9 cutting after transfection of a vector expressing sgCRLF2-1 in a Nalm6 cell line with an integrated Cas9 expression cassette (Nalm6-Cas9). The increase in SUN alone signal indicates a significant drop-off of the TAMRA probe binding the Cas9 cut site (Supplementary Fig. 1A). **E** Quantification of drop-off product in copies/μL as determined by detection of amplicons that are bound by the FAM, TAMRA, Cy5 and/or SUN probes in Nalm6-Cas9 cells following transfection of a vector expressing either sgCRLF2-1 or sgCRLF2-3. Drop-off product of sgCRLF2-1 and sgCRLF2-3 is plotted as mean values ± SEM from 3 biological replicates. Statistical significance was determined by unpaired student's *t* test, with **$p < 0.01$ ($p = 0.0014$ for sgCRLF2-1, $p = 0.0093$ for sgCRLF2-3) ($n = 3$ for sgCRLF2-1, $n = 3$ for sgCRLF2-3). Figures 2A and 2B created with BioRender.com released under a Creative Commons Attribution-NonCommercial-NoDerivs 4.0 International license. Source data are provided as a Source Data file.

File 3, Supplementary Fig. 1A) and were transfected into Nalm6-Cas9 cells (Supplementary Fig. 2). Prior to dPCR, we used High-Throughput rejoin and Genome-Wide Translocation Sequencing (HTGTS-JoinT-seq)[36] to test Nalm6-Cas9 for DSBs genome-wide and to confirm the

specificity of sgRNA cutting. sgIGH-6, targeting the *IGH* locus, was used to induce the bait DSB. The no bait sgRNA controls showed very low numbers of random breakpoint junctions (283 and 161 total breakpoint junctions, Supplementary Fig. 3) while sgIGH-6 transfection resulted in

over a 1000-fold increase in junctions (Supplementary Fig. 3B), but no translocation events. t(X;14) translocation events were only detected after co-transfection with sgCRLF2-1 (Supplementary Fig. 3C). Similar results were obtained with a sgRNA that targets Cas9 to *BCL2* (Supplementary Fig. 3D, E). Thus, untransfected Nalm6-Cas9 cells do not have ongoing genome instability and have robust DSB formation at *CRLF2* and *BCL2* only after sgRNA transfection.

Next, we tested the sensitivity of the dPCR assay to detect mutations at *CRLF2* following DSB formation and repair in our Nalm6-Cas9 cells. sgCRLF2-1 was designed to target Cas9 to sites bound by the TAMRA probe (Supplementary Fig. 1A). dPCR was performed on gDNA from untransfected Nalm6-Cas9 cells or cells transfected with sgCRLF2-1. 2D plots of the dPCR results are shown in Fig. 2C and D. We define a drop-off product as a generated amplicon where one or more of the probes are no longer able to bind. No significant accumulation of the drop-off product is measured in untransfected cells as nearly all amplicons are bound by both TAMRA and the adjacent SUN probes as indicated by the blue dots in the top right quadrant (Fig. 2C). Upon transfection of sgCRLF2-1, there is a dramatic increase in the level of TAMRA drop-off product indicated by the increased number of green dots that signify binding of the SUN probe but not the TAMRA probe (Fig. 2D).

Comparing the copies/μL of drop-off product across multiple replicates demonstrates the high degree of consistency between experiments and shows that transfection of sgCRLF2-1 results in approximately 10% of cells showing TAMRA drop-off, a 1000-fold increase over the baseline levels leading to over 10 copies/μL of drop-off product (Fig. 2E). Interestingly, if we compare this to sgCRLF2-3, which recognizes a site on the opposite DNA strand and was predicted to cut as efficiently as sgCRLF2-1, we measure 5-fold less drop-off product. Thus, dPCR can also provide quantitative information on sgRNA efficiency in Cas9 systems.

### dPCR detects increased indels in response to increased AID expression

To directly correlate AID activity with mutations at ABCs, we integrated a doxycycline (dox) inducible AID expression cassette via lentiviral transduction to generate Nalm6-AID cells. While no protein was detectable with 0 ng/mL of dox, increases in both AID expression (Fig. 3A) and protein levels (Fig. 3B) were evident as dox levels increased. gDNA was prepared from Nalm6-AID cells grown in increasing levels of dox for dPCR analysis of the *CRLF2* ABC (Fig. 3C). With increasing levels of dox, the drop-off products indicating mutations at the FAM and TAMRA bound sites are significantly elevated over the non-AID targeted SUN product. The Cy5 drop-off product is elevated, but not significantly higher than SUN. These results demonstrate that AID preferentially targets the FAM and TAMRA bounds sites, but not the Cy5 site even though it has both an AGCT and a CpG site or the SUN site with no AID-preferred sequence. This is consistent with the AmpSeq data of this region (Fig. 1B) and confirms that not all sites that AID can recognize are accessible to the enzyme[9].

A dPCR assay (Supplementary Fig. 1B) was also designed for the *BCL2* ABC with Cy5 and FAM labeled probes binding at CpG sites and a HEX labeled probe binding at a site not shown to be targeted by AID[13]. A similar dose response correlating increased AID expression with increased drop-off was also measured for the *BCL2* ABC (Fig. 3D). The Cy5 bound site, which showed the most instability via AmpSeq in Nalm6, shows a significant increase in response to dox treatment as does the FAM bound site. This demonstrates that both the *CRLF2* and *BCL2* ABCs are assessable and targeted by AID, but not equally among the CpG or WGCW sites (where W = A or T).

In addition to the dox dose response, we also performed a time-course experiment where AID expression was induced with 1 μg/mL of dox and gDNA was collected over 120 hours. Similar to the dose response, there is a significant increase in accumulation of the FAM

and TAMRA drop-off products that begins approximately 48 hours after AID induction (Fig. 3E). By 72 hours, it appears that all the AID sites that are accessible were targeted as the drop-off accumulation plateaus. Western blots performed with cells induced for 120 hours confirm that AID protein is still present, so the lack of accumulation does not appear to be due to unavailability of AID (Supplementary Fig. 4). Similar to the dose response, neither the Cy5 nor the SUN probes show significant accumulation of drop-off product. The *BCL2* ABC displays a similar significant increase in FAM drop-off product during the same time-course (Fig. 3F).

To confirm AID targeting of the *CRLF2* ABC measured by dPCR, gDNA obtained after the 120-hour time-course was subjected to AmpSeq analysis. By overlaying the AmpSeq data obtained after 120 hours of AID overexpression with the AmpSeq data from Fig. 1B with no detectable AID protein, we can see a clear increase in the number of reads with deletion events due to AID activity that corresponds to the dPCR results (Fig. 3G). The deletion events map at the same region that shows increased deletions in the uninduced cells and correlates to the regions bound by the FAM and TAMRA probes, but not the Cy5 probe, fully consistent with the dPCR results. Interestingly, the deletions following AID seem to spread beyond the major peak after the TAMRA-bound site and taper off as it approaches the Cy5-bound site further supporting that this is a region with low AID accessibility.

### Characterization of Ph-like and Ph+ ALL cohort of patient samples based on genetic admixture and gene expression

To apply our approach to monitor genome instability at *CRLF2* in populations that are susceptible to Ph-like ALL, we leveraged a set of primary samples from the UC Irvine Hematological Malignancies Biorepository. This included a total of 16 ALL patients: 10 Ph-like ALL, 4 Ph+ ALL, and 2 Ph- ALL (Table 1). The majority of these are from patients that self-identified as Hispanic. Additionally, from the UCI Institute for Clinical and Translational Science (ICTS), we obtained samples from 11 healthy donors. 5 self-identified as Hispanic, 4 self-identified as White, and 2 self-identified as Asian (Supplementary Table 1).

Recent analysis of the All of Us genomic data set reported that the concordance between self-reported and predicted race and ethnicity is nearly 90%[37]. Still, the socially constructed label of "Hispanic" can apply to those with genetic similarities to either European or Latin American populations, and studies suggest that the Latin American population is more susceptible to *CRLF2::IGH* translocations that result in Ph-like ALL[24]. Thus, on a subset of the human samples, we used an Illumina global diversity array (GDA) to identify ancestry-specific alleles and compared this to data from the 1000 Genomes Project[38]. All of the self-identified Hispanic patients matched nearly 100% in the broad Admixed American (AMR) category, distinguishing them from the European (EUR) population. As AMR encompasses the entire western hemisphere, the populations were further defined based upon data from the limited, yet more defined, populations within 1000 Genomes (Supplementary Fig. 5). Participants that self-identify as Hispanic mostly have genetic similarities to populations in Mexico, Puerto Rico, Colombia, and Peru, whereas participants that self-identify as White have genetic similarities to populations across Europe.

High *CRLF2* expression is a hallmark of Ph-like ALL but given the high genome instability associated with ALL and that *CRLF2* rearrangements can co-occur in Ph+ ALL cases[39], we wanted to determine *CRLF2* expression levels in all the ALL samples for which we had access. Unfortunately, there were major differences in the quality of patient material obtained that ranged from frozen viable cells from bone marrow aspirates to blood discards that were several days old before arriving in the lab, thus limiting a full characterization of all 16 patient samples. We were able to extract sufficient RNA from 12 patient

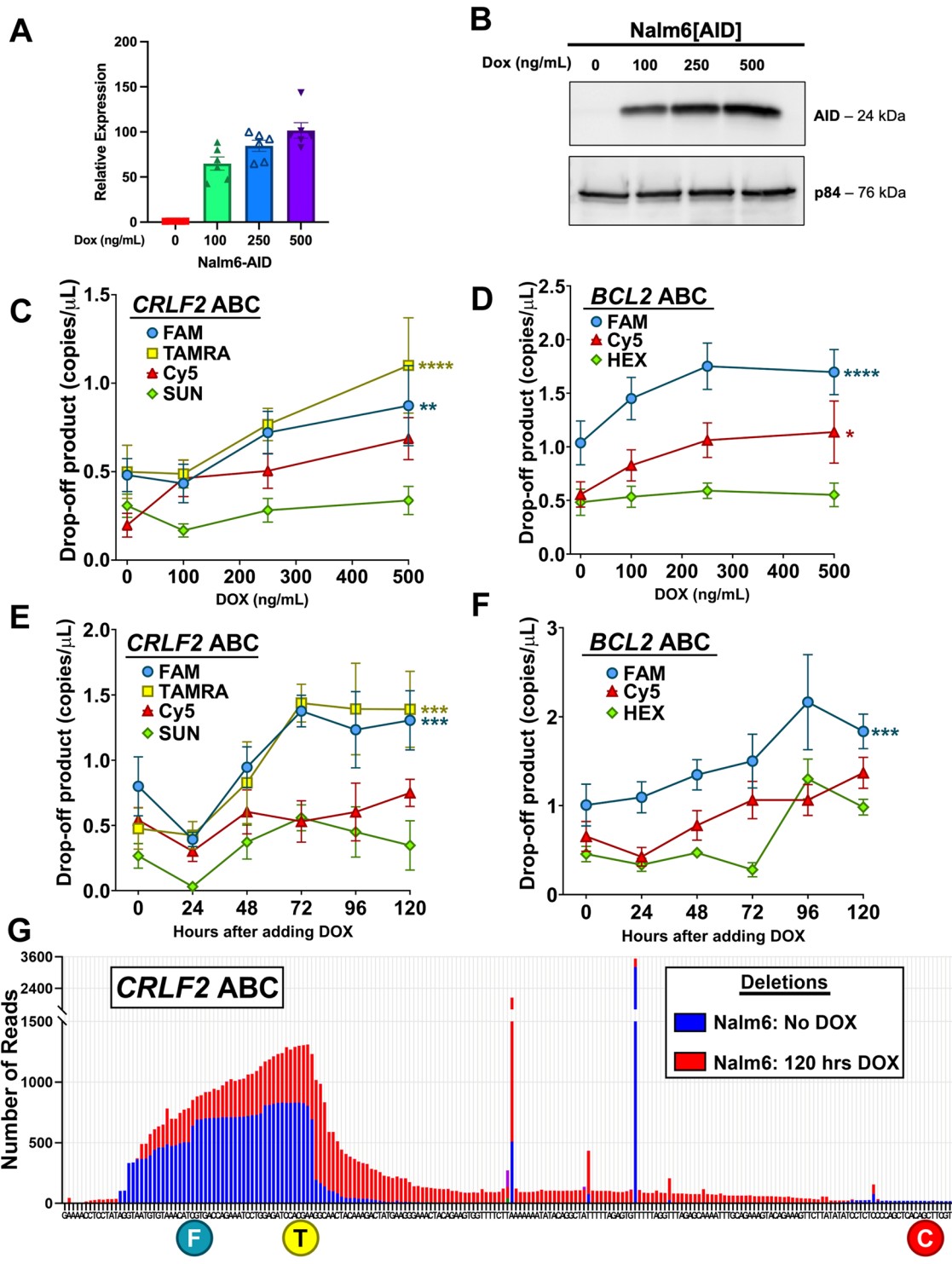

samples. Among Hispanics, this consisted of 5 Ph-like, 2 Ph+, and 2 Ph-ALL patients. Additionally, we also included 3 Asian ALL patients (2 Ph+ and 1 Ph-like) to provide a baseline of expression in a background with very low *CRLF2*-linked Ph-like ALL risk[23].

As expected, Ph-like-1 to Ph-like-4 and Ph-like-6 all displayed elevated levels of *CRLF2* expression (Fig. 4A), as this was the basis of their diagnosis. Though the levels of *CRLF2* expression in Ph-Like-1 and Ph-Like-2 were clearly hyper-elevated in relation to the other patient samples. Interestingly, Ph + −2 and ALL-1 also showed high levels of *CRLF2* expression. The Asian Ph + −3 and Ph + −4 samples had low *CRLF2* expression while the Asian Ph-like-9 showed expression consistent with other Ph-like patients. The variability of *CRLF2* expression

in the Ph-like patients and the high expression in Ph + −2 and ALL-1 further emphasizes that additional diagnostic criteria may be necessary for an accurate diagnosis.

Since we expect that high levels of AID contribute to a subset of Ph-like ALL, we also measured expression of the *AICDA* gene encoding AID (Fig. 4B). Ph-like-1 shows a dramatically increased level of relative *AICDA* expression, as does Ph-like-6. Ph-like-2, −3, and −4 show moderate to little increase in expression. Surprisingly, Ph + −2 also shows high levels of *AICDA* expression, in addition to high levels of *CRLF2* expression (Fig. 4A). Neither of the genes would be expected to have high expression in a Ph+ patient, which is why we further characterize this sample below. One possibility *AICDA* was not consistently detected

**Fig. 3 | dPCR detects increased indels in response to increased AID expression in human pre-B cell lines. A** Relative expression of the *AICDA* gene encoding AID in Nalm6 cells with a lentivirally-integrated dox-inducible AID cassette determined by qPCR in cells cultured with indicated amounts of doxycycline. Relative expression data is plotted as mean values ± SEM from 6 biological replicates ($n = 6$). **B** Western blot showing protein abundance of AID in dox-induced Nalm6-AID cells relative to p84. **C** Quantification of dPCR drop-off products for the FAM, TAMRA, Cy5 and SUN probes associated with the *CRLF2* ABC site using gDNA harvested from Nalm6-AID cells induced with indicated concentrations of doxycycline. Drop-off product data is plotted as mean values ± SEM from 3 biological replicates. Statistical significance was determined by two-way ANOVA with Dunnett's multiple comparisons test to compare to the SUN drop-off values.**$p < 0.01$ and ****$p < 0.0001$ ($p = 0.0012$ for FAM, $p < 0.0001$ for TAMRA)($n = 9$ for FAM, TAMRA, Cy5 and SUN). **D** Quantification of dPCR drop-off products for the FAM, Cy5 and HEX probes associated with the *BCL2* ABC using gDNA harvested from Nalm6-AID cells induced with the indicated concentrations of doxycycline. Drop-off product data is plotted as mean values ± SEM from 3 biological replicates. Statistical significance was determined by two-way ANOVA with Dunnett's multiple comparisons test to compare to the HEX drop-off values. *$p < 0.05$ and

****$p < 0.0001$ ($p < 0.0001$ for FAM, $p = 0.0369$ for Cy5)($n = 8$ for FAM, Cy5 and HEX). **E** Quantification of dPCR drop-off products for the FAM, TAMRA, Cy5 and SUN probes associated with the *CRLF2* ABC site using gDNA harvested every 24 hours from Nalm6-AID cells induced with 1 μg/mL of doxycycline over a 120-hour time period. Drop-off product data is plotted as mean values ± SEM from 4 biological replicates. Statistical significance was determined by two-way ANOVA with Dunnett's multiple comparisons test to compare to the SUN drop-off values. ***$p < 0.001$ ($p = 0.0003$ for FAM, $p = 0.0003$ for TAMRA) ($n = 4$ for FAM, TAMRA, Cy5 and SUN). **F** Quantification of dPCR drop-off products for the FAM, Cy5 and HEX probes associated with the *BCL2* ABC site using gDNA harvested every 24 hours from Nalm6-AID cells induced with 1 μg/mL of doxycycline over a 120-hour time period. Drop-off product data is plotted as mean values ± SEM from 4 biological replicates. Statistical significance was determined by two-way ANOVA with Dunnett's multiple comparisons test to compare to the HEX drop-off values. ***$p < 0.001$ ($p = 0.0006$) ($n = 4$ for FAM, Cy5 and HEX). **G** Comparison of AmpSeq indel events in Nalm6 cells vs. Nalm6-AID cells induced with 1 μg/mL of doxycycline for 120 hours. The two stand-alone peaks in this region are by-products of amplicon generation as they map to a run of A's and a run T's and are not due to AID. Source data are provided as a Source Data file.

## Table 1 | De-Identified Material Collected from ALL Patients Treated at the University of California, Irvine

| Sample | Age Range | Sex | Ethnicity | Genetics/FISH Findings Immunophenotype | Source | % Blasts |
|---|---|---|---|---|---|---|
| Ph-Like ALL[1] | | | | | | |
| Ph-like-1 | 50-80 | M | Hispanic | CRLF2 and IGH Rearrangement[2]<br>CD10+, CD19+, CD20+, CD34 (subset), CD38+, cCD22+, cCD79a, and cTDT+ | BM | 95 |
| Ph-like-2 | 20-50 | M | Hispanic | CRLF2 and IGH Rearrangement[2]<br>CD19+, CD10+, CD20+, cCD79a+, TdT+, CD34 (subset), and HLA-DR (subset) | BM | 95 |
| Ph-like-3 | 50-80 | F | Hispanic | CRLF2 and IGH Rearrangement[2]<br>CD34+, cTdT+, CD10+, CD19+, CD22(subset), CD33+, cCD79a+, cCRLF2+, HLA-DR+, CD38(subset) | BM | 95 |
| Ph-like-4 | 50-80 | M | Hispanic | CRLF2::IGH[3]<br>CD19+, CD10+, CD22 (SUBSET), CD34+, HLA-DR+, CD33 (SUBSET), CD20- | PB | 90 |
| Ph-like-5 | 20-50 | F | Hispanic | P2RY8 and CRLF2 Rearrangement[2]<br>CD19 + CD34 + CD10(subset), CD22(very dim), CD38 + HLA-DR+ | BM | 62 |
| Ph-like-6 | 20-50 | F | Hispanic | P2RY8 and CRLF2 Rearrangement[2]<br>CD19 + CD34 + CD10(very small subset), CD22(subset), CD38(dim), CD11b+, cCD79a(dim), cTdT(dim), cCRLF2(dim), HLA-DR+ | BM | 50 |
| Ph-like-7 | 50-80 | M | Hispanic | P2RY8::CRLF2[4]<br>CD19+, CD10+, CD20+, CD22+, TdT (subset), CD38+, CD34 (large subset), CD79a+, HLA-DR+ | PB | 59 |
| Ph-like-8 | 20-50 | M | Hispanic | P2RY8 and CRLF2 Rearrangement[2]<br>CD10+, CD19+, CD20+, CD22+, cCRLF2+, cCD79a+, CD34+, CD38+, HLA-DR+, and TdT+ | PB | 90 |
| Ph-like-9 | 20-50 | M | Asian | CRLF2 and IGH Rearrangement[2]<br>CD19 + CD34 + CD10+, CD22+, CD38 + HLA-DR, aberrant expression of CD33(subset) | BM | 56 |
| Ph-like-10 | 50-80 | M | White | CRLF2 and IGH Rearrangement[2]<br>CD34+ TdT(large subset), CD19+, cCD79a+, CD10+, CD10+, CD20(subset, 45%), sCD22(subset), cCRLF2+, CD38+, HLA-DR+ | PB | 91 |
| Ph-Positive ALL | | | | | | |
| Ph + −1 | 50-80 | M | Hispanic | BCR::ABL1<br>CD19+, CD10+, CD34+, CD79a (cytoplasmic) and TDT (cytoplasmic) with expression of a myeloid marker of CD13 | BM | 95 |
| Ph + −2 | 20-50 | M | Hispanic | BCR::ABL1<br>CD19+, CD10+, CD34+, HLA-DR+, CD33+, CD38+, cCD79a+, and cTDT+ | BM | 90 |
| Ph + −3 | 50-80 | F | Asian | BCR::ABL1<br>CD34+, CD10+, CD19+, CD79a+ and<br>TdT+ CD22(subset) | BM | 90 |
| Ph + −4 | 50-80 | F | Asian | BCR::ABL1<br>CD19+, CD10+, CD22 (subset), CD34+, HLA-DR+, CD33 (subset), CD20- | BM | 90 |
| Ph-Negative ALL | | | | | | |
| ALL-1 | 20-50 | M | Hispanic | Negative for BCR::ABL1, IGH, CRLF2, MYC, and KMT2A Rearrangements<br>CD10+, CD20+, CD79a+, TdT+ | PB | 73 |
| ALL-2 | 50-80 | M | Hispanic | KMT2A::AFF1<br>partial CD33+, CD34+, CD38+, CD79a+, HLA-DR+ TdT+ | BM | 80 |

[1]Ph-like ALL was diagnosed using flow cytometry to detect increased CRLF2 expression.
[2]Break-apart probes were used to detect *CRLF2* and *IGH* rearrangements.
[3]Confirmed by long-read sequencing in this work (see Fig. 5C).
[4]Next generation sequencing was used to detect a *P2RY8::CRLF2* fusion RNA transcript.

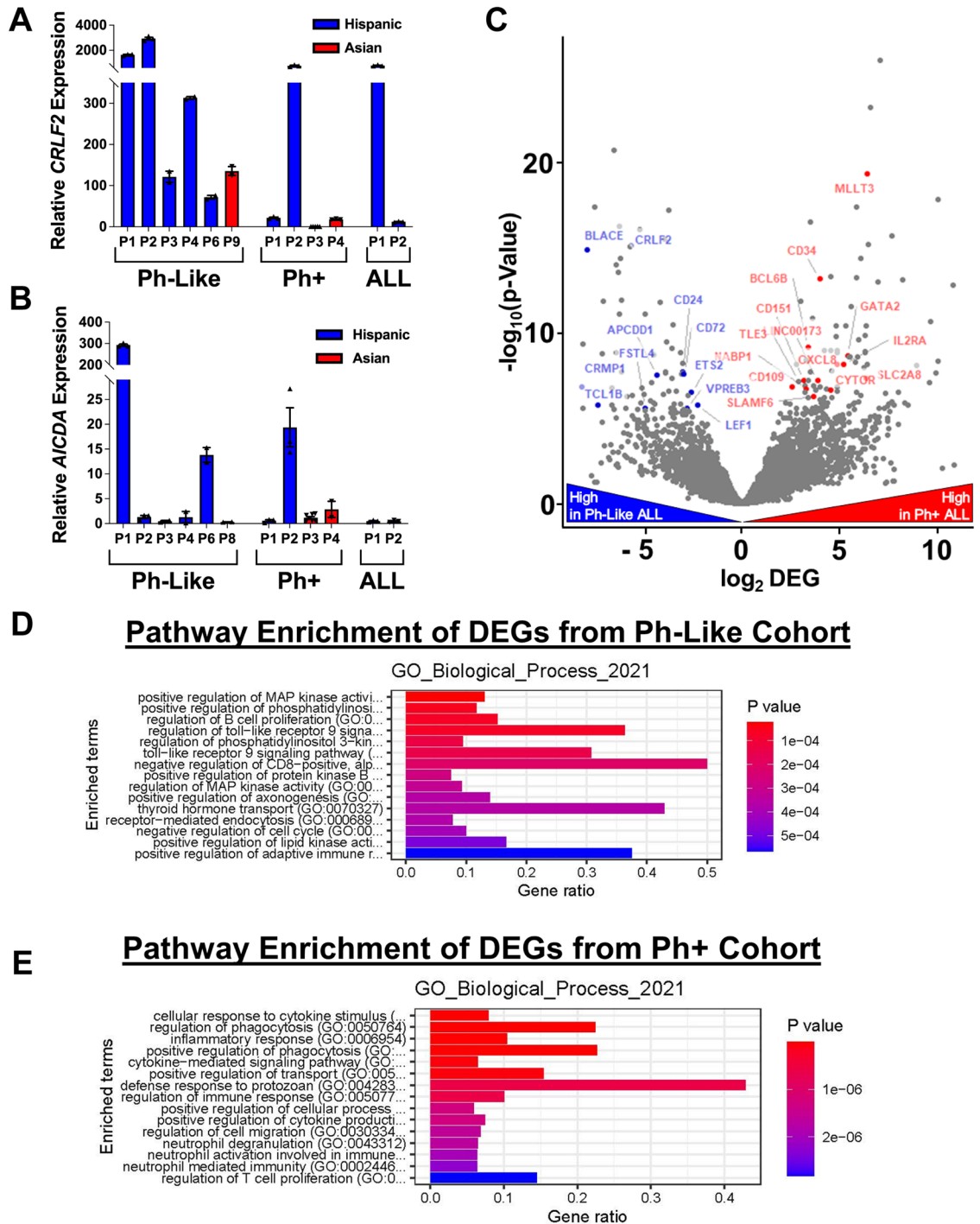

**Fig. 4 | Expression analysis on Ph-like, Ph-, and Ph+ ALL cohorts shows detection of variable *CRLF2* and AID expression levels in LA patients. (A)** Relative expression of *CRLF2* from the indicated patient samples. Blue are Hispanic patients diagnosed with the indicated ALL subtype and red are Asian patients diagnosed with the indicated ALL subtype. Relative expression is plotted as mean values ± SEM from technical replicates of each individual patients (n = 5 for Ph + : P3) (n = 3 for Ph-Like: P1, P2, Ph+: P1, P2, P4, ALL: P1, P2) (n = 2 for Ph-Like: P3, P4, P6, P9). **B** Relative expression of *AICDA* (gene encoding AID) from the indicated patient samples. Blue are Hispanic patients diagnosed with the indicated ALL subtype and red are Asian patients diagnosed with the indicated ALL subtype. Relative expression is plotted as mean values ± SEM from technical replicates of each individual patient (n = 7 for Ph+: P3) (n = 3 for Ph-Like: P1, P2, Ph+: P1, P2, ALL: P1, P2) (n = 2 for Ph-Like: P3, P4, P6, P8, Ph+: P4.) **C** Volcano plot depicting differently expressed genes (DEGs) between Ph-like-1 and Ph-like-2 (blue) versus Ph+1 and Ph+2 (red). A complete list of DEGs are shown in Supplementary Data 2. Significance is calculated using a Wald t-test. Gene Set Enrichment Analysis (GSEA) was performed using the GO Biological Process pathway database showing pathway enrichment from the **D** Ph-like −1 and Ph-like-2 and **E** patients Ph + −1 and Ph+2. Source data are provided as a Source Data file.

in Ph-like ALL samples may be because we are looking at samples collected from patients at a single timepoint. Increased *AICDA* expression in pre-B cells may be transient, as has been suggested for other APOBEC family members[40]. This emphasizes the importance of

detecting mutation scars from prior AID activity to fully determine if it is involved in disease etiology.

Next, to compare global gene expression changes between Ph-like and Ph+ ALL patients, we performed RNA-seq and compared Hispanic

patients Ph-like-1 and Ph-like-2 to Hispanic patients Ph + −1 and Ph + −2 with analysis of differentially expressed genes (DEG) showing a number of differences in the expression pattern (Fig. 4B). As expected, *CRLF2* expression is significantly higher in all the Ph-like samples. A number of the DEGs in this comparison have not previously been identified as mutated in Ph-like ALL, but we have highlighted the ones that may be involved in cancer development. A complete list of all genes that showed differential expression can be found in Supplementary Data 2.

We further highlight the differences between the two cohorts by performing Gene Set Enrichment Analysis (GSEA) to determine pathway enrichment of DEGs using the Gene Ontology Biological Process database (Figs. 4D, C) as well as the Reactome, and Drug Signatures Database (DsigSB) databases (Supplementary Fig. 6). The analysis reveals very little overlap between enrichment in the Ph-like and Ph+ patients, confirming that there are major differences between these ALL subtypes. Several of the most significant pathways affected in Ph-like patients include those that regulate kinase activation, which aligns with previous reports that distinguish *CRLF2*-linked ALL from Ph+ ALL. Utilization of multiple kinase pathways may make Ph-like ALL more refractory to treatments that only target a single kinase pathway[41–43].

### dPCR of Hispanic patient samples shows increased instability at *CRLF2*

As dPCR uses only a fraction of the DNA used for high-throughput sequencing and provides absolute quantification, there would be a clear advantage to using this method to detect instability at ABC sites like *CRLF2*. Of the 8 Hispanic Ph-like ALL samples we obtained, 4 have *CRLF2* and *IGH* rearrangements (Ph-like-1-4) and 4 have *CRLF2* and *P2RY8* rearrangements (Ph-like-5-8) (Table 1). *CRLF2* rearrangements resulting in a *CRLF2::IGH* translocation are thought to be driven by AID activity while those leading to the *P2RY8::CRLF2* as a result of an intrachromosomal deletion are not[10,14]. We compared drop-off product levels from Ph-like-1-3 (Ph-like-4 was excluded as described below) and Ph-like-5-8 and indeed found that there were higher levels of drop-off in the *CRLF2::IGH* samples that indicate mutations at AID-target sites (Fig. 5A). For the FAM and Cy5 probes, this was significantly higher than in the *P2RY8::IGH* samples, further supporting increased AID activity at *CRLF2* in these Ph-like ALL patients. In contrast, we see no significant differences in drop-off products at the *BCL2* locus in these patients (Supplementary Fig. 7). Even though we detected drop-off at *BCL2* in Nalm6 cells, perhaps this locus was not accessible to AID to a sufficient degree in the patient samples.

Of the samples tested, two had patterns of drop-off product at *CRLF2* that differed dramatically from the others. Both Ph-like-4 and Ph + −2 showed a significant increase in the SUN drop-off product, but not the other three probes (Fig. 5B). This would indicate that a portion of one of the chromosomes was lost or mutated. For the Ph-Like-4 sample, it would make sense that formation of a *CRLF2::IGH* translocation could lead to some loss of DNA information, even though it is reciprocal. The *BCR::ABL1* translocation in Ph + −2 is not expected to involve *CRLF2*, but cases of co-occurrence have been reported[39] and this sample did present with high *CRLF2* and *AICDA* expression (Figs. 4A, B). To further investigate if this dPCR pattern is indicative of a subset of *CRLF2* rearrangements, we performed whole genome long-read sequencing on these two patients.

Analysis shows that patient Ph-Like-4 has a number of rearrangements (Fig. 5C). Importantly, the break-apart FISH probes used to diagnose Ph-Like ALL do not fully confirm translocations, but sequencing results clearly and definitively demonstrate the presence of the t(X;14) *CRLF2::IGH* translocation (Table 1). Several split reads from the sequence alignment also indicate structural variants (SVs) involving chromosome 9 and chromosomes 22 and 1, between chromosomes 7 and 18, and between chromosomes 14 and 16 shown as connected lines on a circle plot (Fig. 5C). There is also increased indel

formation at the loci involved in translocations, represented by a vertical unconnected line, that indicates multiple DSB and rejoin events occurred at these sites. Several of these SVs correspond to loci known to be mutated in Ph-like ALL including *JAK2, IKZF1, TCR,* and *CDKN2A*[24,44–47]. Ph-like-4 has 2 SVs at the *JAK2* locus, 2 SVs at the *IKZF1* locus, and SVs at the *CDKN2A* and *TCR* loci (Supplementary Table 2). As we continue to collect and sequence patient samples, it will be interesting to determine if this level of instability is common in Ph-like ALL.

For the Ph + −2 sample, we unfortunately did not have adequate DNA from this patient to reach the same read depth (Supplementary Fig. 8), but there are several notable features from this analysis. For one, the expected *BCR::ABL1* translocation is clearly present (Fig. 5D). Overall, there are far fewer SVs in Ph + −2 compared to Ph-like-4 (Supplementary Table 2). Also, instability on chromosome 14 in Ph + −2 is consistent with RAG activity resulting in rearrangements at the *IGH* locus and may be a distinguishing feature between Ph + -ALL and Ph+ myeloid leukemias[48]. Interestingly, we do not see instability at the *CRLF2* locus on the X chromosome even though dPCR indicates a deletion or mutation at this site (Fig. 5B) and qPCR shows high *CRLF2* (Fig. 4A) and *AICDA* (Fig. 4B) expression. If we directly compare the read alignments, we can see instability at the *IGH* locus on chromosome 14 in Ph + −2 that is consistent with translocation breakpoints mapped in Ph-like-4 (Supplementary Fig. 8A). Consistent breakpoints are also present on chromosome X (Supplementary Fig. 8B), but the lack of read depth prevents confirmation. Thus, based on this evidence, we are hesitant to rule out that a *CRLF2* rearrangement is present. A complete listing of annotated variants is shown in Supplementary Table 2 for Ph-like-4 and Ph + −2.

### CD19+ B cells in healthy Hispanic donors have increased instability at the *CRLF2* ABC

If increased AID activity is resulting in *CRLF2* instability and driving *CRLF2::IGH* translocations, we also wanted to know if this is detected in samples from healthy, cancer-free donors (Supplementary Table 1) as this may be a prognostic feature of Ph-like ALL. From healthy donors, we were able to obtain sufficient fresh whole blood samples to prepare gDNA from the CD19- cells and the CD19+ B cell population so that we could directly compare a population of cells that had AID expression during their maturation to those that did not. We were also able to further compare 5 Hispanic to 6 non-Hispanic donors.

Strikingly, we see increased *CRLF2* mutations at all 4 probe binding sites in the CD19+, but not the CD19-, cell populations (Fig. 5E). Moreover, while elevated in both populations, significant increases in the FAM and SUN drop-off products are only measured in the Hispanic and not the non-Hispanic samples. Taking the same samples and instead interrogating the *BCL2* ABC, we see a starkly different pattern. In both the Hispanic and non-Hispanic samples, the level of drop-off is comparable between CD19- and CD19+ cells (Fig. 5F). The exception to this is that the CD19+ cells have a higher level of Cy5 drop-off, but only in the Hispanic samples. These data suggest that AID-induced instability at *CRLF2* may be a general feature during B cell maturation, yet the higher instability in Hispanics may be driving the disparity in Ph-like ALL with *CRLF2::IGH* rearrangements.

## Discussion

Our major goal is to understand the etiology of several types of B cell malignancies that are related by the fact that they carry a chromosomal translocation that occurs at the pro-B/pre-B cell stage of development based on translocation junction analysis[13,14]. Two such rearrangements are shown in Fig. 1. The *CRLF2::IGH* translocation presents in Ph-like ALL, a disease of immature B cells, and the rearrangement involves a DSB occurring between D and J cassettes of the *IGH* locus that can only consistently occur through activity of the RAG complex. The *BCL2::IGH* translocation occurs in over 85% of follicular lymphoma and 20% of diffuse large B-cell lymphoma[49], which are both diseases of mature B

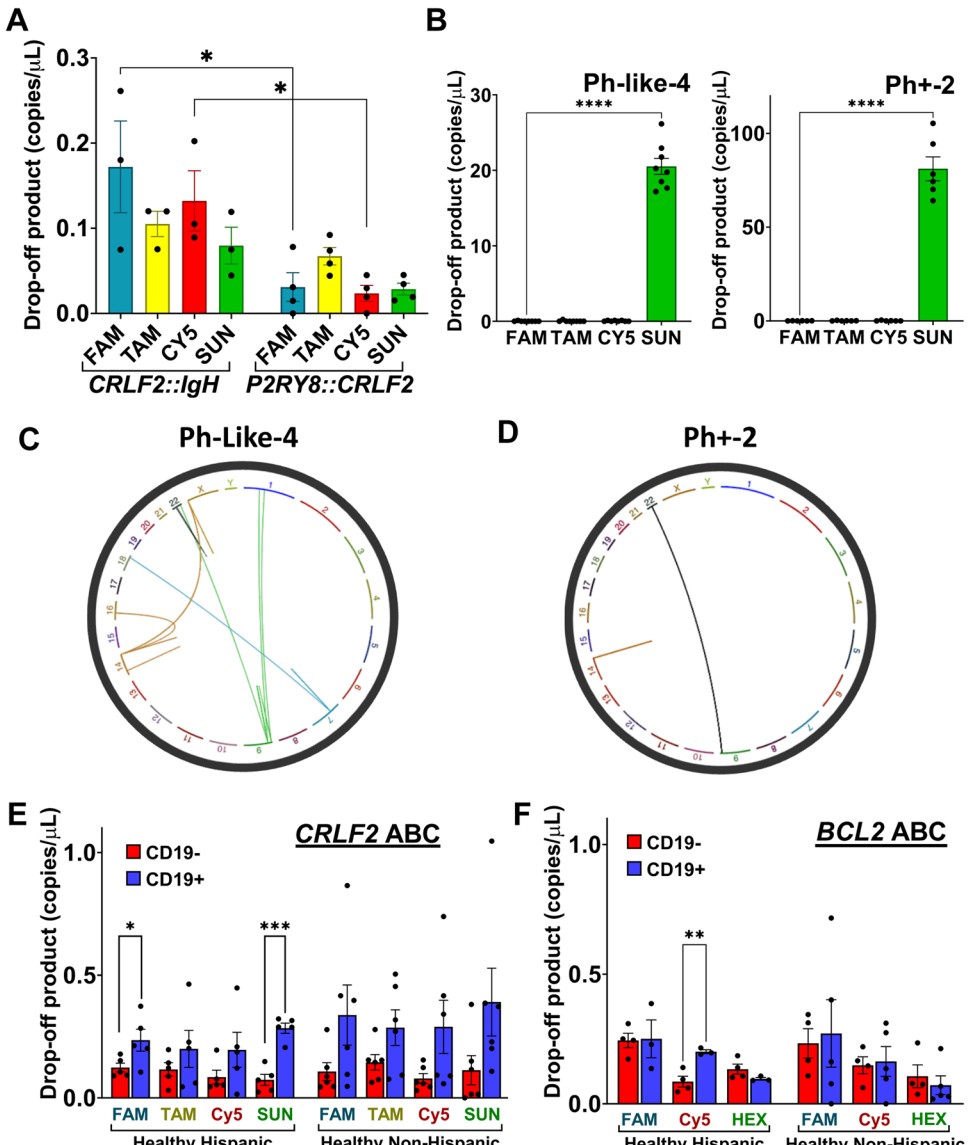

**Fig. 5 | dPCR of Hispanic Ph-like ALL patients and healthy donors shows increased mutation at *CRLF2*. A** Comparison of quantified drop-off product for the FAM, TAMRA, Cy5 and SUN probes associated with the *CRLF2* ABC between Hispanic Ph-Like ALL patients with *CRLF2::IGH* or *P2RY8::CRLF2* rearrangements. Each data point represents the mean from at least 6 technical replicates using gDNA from an individual patient. Drop-off product for *CRLF2::IgH* is plotted as mean values ± SEM from 3 individual diagnosed patients. Drop-off product for *P2RY8::CRLF2* is plotted as mean values ± SEM from 4 individual diagnosed patients. Statistical significance was determined by unpaired student's *t* test, with \**p* < 0.05 (*p* = 0.0354 for FAM, *p* = 0.0185 for Cy5) (*n* = 3 for *CRLF2::IgH*, *n* = 4 for *P2RY8::CRLF2*). **B** Individual drop-off product for the FAM, TAMRA, Cy5 and SUN probes associated with the *CRLF2* ABC in Ph-like-4 and Ph + −2 patients. Drop-off product for Ph-like-4 is plotted as mean values ± SEM from 8 technical replicates. Drop-off product for Ph + −2 is plotted as mean values ± SEM from 6 technical replicates. Statistical significance was determined by unpaired student's *t* test, with \*\*\*\**p* < 0.0001 (*p* < 0.0001 for Ph-Like−4, *p* < 0.0001 for Ph + −2) (*n* = 8 for Ph-Like-4, *n* = 6 for Ph + −2). **C** Circular view plot generated from long read whole genome

sequencing of patient Ph-Like−4 and **D** patient Ph + −2 using the web-based Integrated Genomics Viewer (IGV) program[70]. Connected lines show rearrangements involving two chromosomes and vertical lines show instability at the locus of one chromosome. Quantified drop-off product comparison between CD19- peripheral blood mononuclear cells (PBMC's) and isolated CD19+ cells in healthy Hispanics donors versus healthy non-Hispanic donors in the **E** *CRLF2* ABC and the **F** *BCL2* ABC. Each data point represents the mean from at least 6 technical replicates using gDNA from an individual patient. Drop-off product for *CRLF2* ABC (**E**) is plotted as mean values ± SEM from individual healthy Hispanic and non-Hispanic donors. Statistical significance was determined by unpaired student's *t* test, with \**p* < 0.05 and \*\*\**p* < 0.001 (*p* = 0.0492 for FAM, *p* = 0.0001 for SUN) (*n* = 5 for Healthy Hispanic, *n* = 6 for Healthy Non-Hispanic). Drop-off product for *BCL2* ABC (**F**) is plotted as mean values ± SEM from individual healthy Hispanic and non-Hispanic donors. Statistical significance was determined by unpaired student's *t* test, with \*\**p* < 0.01 (*p* = 0.0061 for Cy5) (*n* = 4 for CD19- from Healthy Hispanic, *n* = 3 for CD19+ from Healthy Hispanics, *n* = 4 for CD19- from Healthy Non-Hispanic, *n* = 5 for CD19+ from Healthy Non-Hispanics). Source data are provided as a Source Data file.

cells. Yet, the translocation junction again shows that the *IGH* DSB was initiated by the RAG complex, thus indicating that the translocation is of pre-B cell origin. B cells harboring this translocation continue to mature and acquire additional mutations until it fully transforms into a mature B cell lymphoma[50]. This evokes a model where high levels of RAG activity in pre-B cells combined with aberrant AID activity at ABCs

generates the two DSBs required for a cancer initiating chromosomal translocation (Fig. 6). Through our cell-based model, we were able to clearly demonstrate a dose response where, as AID levels increase, the ABCs at both *CRLF2* and *BCL2* are subjected to increased genome instability. This was not true for all sites that harbor AID-preferred target sequences, such as the region bound by the Cy5 probe at *CRLF2*.

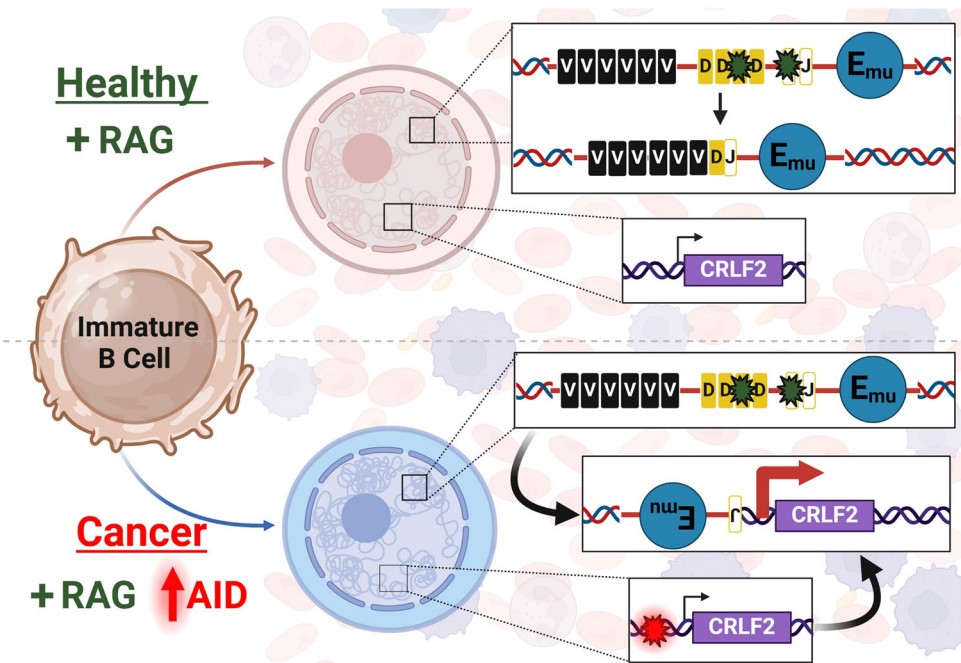

**Fig. 6 | Model of chromosomal translocation formation between the *IGH* and *CRLF2* loci in B cell Ph-like ALL.** Healthy pro-B/pre-B cells express high levels of the Recombination Activating Gene (RAG) complex to carry out V(D)J recombination to maintain functional adaptive immunity. First D and J regions are combined as shown in the top panel. In subsequent steps, RAG induced DSBs join V to DJ regions (not shown). High levels of *IGH* transcription in B cells are maintained by the presence of the mu enhancer ($E_{mu}$) located between the VDJ and constant region gene cassettes of *IGH* (for simplicity, the constant region cassettes are not shown). The bottom panel shows that cells are transformed into a cancer state when RAG and AID are expressed simultaneously to generate RAG-induced DSBs at *IGH* and AID-induced DSBs at ABC sites such as *CRLF2*. Aberrant repair of these DSBs results in a chromosomal translocation placing $E_{mu}$ upstream of *CRLF2* leading to over-expression that puts the cells into a hyperproliferative state. Figure 6 created with BioRender.com released under a Creative Commons Attribution-NonCommercial-NoDerivs 4.0 International license.

Our study has also allowed us to better understand the Ph-like ALL health disparity within Hispanic populations with Latin American ancestry. Using gDNA from patient samples, we demonstrated that Hispanic patients diagnosed with Ph-like ALL that have *CRLF2* and *IGH* rearrangements had significantly higher levels of genome instability at *CRLF2* that was not present in the Ph-like ALL patients with *P2RY8* and *CRLF2* rearrangements, underscoring the different origins of these two rearrangements. Furthermore, healthy Hispanic donors had a significantly higher level of instability at *CRLF2* in their CD19+ B cell population (Fig. 5E). These data suggest that *CRLF2* is vulnerable to AID damage during B cell maturation. This increased damage at *CRLF2* may potentiate translocations with the *IGH* locus during a failure to efficiently repair RAG-induced DSBs during V(D)J recombination. If this is the case, further examination of what may be affecting the repair kinetics of the RAG-induced DSB at *IGH* is warranted to fully understand the translocation mechanism. It is important to note that confirmation of these results will need a larger pool of Hispanic and Latino Ph-like ALL patients beyond the limited number we were able to collect here and is a central goal of our future work.

Interestingly, the *BCL2* locus was not targeted in either the Ph-like ALL patients or the healthy donors. (Fig. 5F, Supplementary Fig. 7). On the one hand, this makes sense as follicular lymphoma is not associated with a Hispanic health disparity, but it raises a critical question as to why *BCL2* can be targeted in the Nalm6 cell line, but not in the patient samples. A strong possibility is that factors beyond just the primary DNA sequence dictate sensitivity to AID. This has already been demonstrated by the dPCR performed in Nalm6 cells as not all AID target sequences were equally hit (Fig. 3). DNA methylation can affect the accessibility and vulnerability of DNA sequences to AID, and previous work has shown variable or partial methylation in ABC sites including *CRLF2* and *BCL2*[9,10,51,52]. Thus, interrogating the methylation status of the *CRLF2* and other ABCs will be critical to fulling understanding how AID can target regions and lead to oncogenic translocations.

Diagnosis of Ph-like ALL has remained difficult due to the lack of agreement on what type of gene expression profile or rearrangement constitutes Ph-like ALL, the heterogeneity of mutations and rearrangements in patients, and the lack of concordance regarding rearrangements in the United States versus Europe due to differences in genetic ancestry[24]. Currently several different assays can be conducted to confirm a Ph-like ALL diagnosis. Following an initial test using RNA-seq for detection of fusion transcripts and aberrantly expressed kinases, additional methods can be used to determine specific translocations and mutations including flow cytometry for increased *CRLF2* receptors at the cell surface and FISH analysis of cytogenetically cryptic rearrangements[53]. Despite evidence that AID is involved in the DSB leading to the *CRLF2::IGH* translocation, increased AID activity is not currently examined in any of these diagnostic assays. Likely, this is because increased AID levels occur transiently following exposure to external stimuli, and do not remain high in cells throughout the course of the disease. Our own qPCR data shows elevated AID expression in Ph-like patients, but it is not consistently high or associated with the *CRLF2::IGH* class. Still, as we demonstrate here, a peak of AID activity in immature B cells can increase mutations and indels at ABCs in pre-B cells and that evidence of AID activity may remain as information scars in cancerous or pre-cancerous B cells in human patients. Future studies will further add to our cohort of Hispanic Ph-like ALL patients to determine if AID signatures are a consistent feature of cancers linked to ABC sites. COSMIC signatures indicative of AID activity include SBS84 and SBS85[54], but determining this signature in a patient requires whole genome sequencing with a matched normal sample, which is cost prohibitive. dPCR may offer a rapid and cost-effective alternative

to measure instability at sites proposed to be involved in cancer etiology that are sensitive to aberrant AID activity.

Overall, this analysis fills a critical gap between developing genetic assays in cells to monitor DSB formation and repair via expression of a selectable or detectable marker and genome-wide sequencing of primary or patient material to measure genome instability at a specific locus. In addition to using drop-off assays to examine mutations created at ABC sites as an indicator of instability caused by aberrant AID levels in pre-B cells, allele specific probes can also be designed for mutations associated with cancer subtypes. As demonstrated by the long-read sequencing of patient Ph-like-4 (Fig. 5C), Ph-like ALL often has mutations in *JAK2* and *IKZF1*, particularly in cases where a *CRLF2* rearrangement is also present[24,44]. These additional mutations can indicate a more aggressive cancer that may be responsive to therapies other than the standard of care[55] making it critical to identify these mutations early.

As the cancer field continues to move towards personalized medicine approaches to treatment, it is going to be important to not only take a holistic approach to studying cancer genomes that incorporate multiple races and ethnicities, but also to develop rapid and economical approaches to assessing cancer risk that are widely available to people independent of socioeconomical stratifications. We are currently lagging in both of these goals. Extensive analysis of a larger cohort of Hispanic patients will allow us to develop diagnostics that can indicate risk for diseases such as Ph-like ALL and determine how factors such as genetic ancestry may be driving cancer health disparities and aid in the discovery of new therapies.

## Methods

### Cell lines and culture conditions
Nalm6 (CRL-3273) and Reh (CRL-8286) cell lines were purchased from the American Type Culture Collection (ATCC, Manassas, Virginia) and grown in RPMI-1640 supplemented with 10% FBS, 1% Penicillin/Streptomycin, 1% Glutamax (Gibco, cat# 35050-061), and 50 µM β-mercaptoethanol (Sigma, cat# M7522-100ML). Cell lines with the transduced AID doxycycline (dox)-inducible cassette were grown under G418 selection (1 mg/mL of G418) (Gold Biotechnology, cat# G-418-10). Cells with constitutive Cas9 expression were grown in complete media further supplemented with 0.75 µg/mL of Puromycin (Gold Biotechnology, cat# P-600-100). All cell lines stated above were grown at 37 °C and 5% CO$_2$.

### Vector construction
Single guide RNAs (sgRNAs) targeting ABCs were designed using Benchling. Synthesized DNA oligos (Integrated DNA Technologies (IDT), Coralville, IA) were ligated into the *Bbs*I sites of pSPgRNA[56]. Gateway cloning was used to generate pINDUCER-AID using the Invitrogen Clonase II (cat# 11791-020, cat# 11789-020) kit to integrate AID into pINDUCER20[57]. Nucleofections were performed using the Amaxa Biosystems Nucleofector II with program T-001.

### Construction of cells with dox-inducible AID or constitutive Cas9 cassettes
Virions with pINDUCER-AID were generated using a 3$^{rd}$ generation lentiviral system in HEK293T cells[58]. For constitutive Cas9, the lentiCRISPRv2[59] vector was used. Nalm6 cell lines with the dox-inducible AID cassette (hereon referred to as Nalm6-AID) were generated by transduction and selection in complete media supplemented with 1 mg/mL of G418. A similar process was done using virions that incorporate the Cas9 expression cassette and create Nalm6-Cas9 with selection done using 0.75 µg/mL of puromycin.

### Patient Samples
We have complied with all relevant ethical regulations. Patient samples were obtained by informed consent through Institutional Review Board-approved protocols of the University of California, Irvine (IRB#'s: 634, HS2012-8716, 2014-1709). The study was conducted in accordance to the criteria set by the Declaration of Helsinki. No sex or gender analysis was performed as ALL does not show a sex bias and all samples that were available were utilized. Samples were obtained from bone marrow or blood (if bone marrow was not available and blasts were >50% in peripheral blood). De-identified healthy donor blood samples were provided by the UCI Institute for Clinical and Translational Science. Mononuclear cells were obtained using Lymphoprep (cat# 07851) and SepMate tubes (cat# 85450) (STEMCELL Technologies, Vancouver, British Columbia). CD19 + B cells were isolated using the EasySep Human CD19 Positive Selection Kit II (STEMCELL Technologies, cat#17814) with the remaining CD19- cells also preserved. Cells were used fresh or thawed from frozen cells preserved in 90% Fetal Bovine Serum (FBS) + 10% DMSO.

### dPCR
dPCR samples were prepared as described previously[60]. Briefly, 9 µL reactions contained 1x Absolute Q DNA dPCR Master Mix (Thermo Fisher Scientific, cat# A52490), 900 nM of each forward and reverse amplicon primer, 250 nM of FAM, TAMRA, Cy5 and/or SUN/HEX TaqMan probes, 1 unit of *Avr*II restriction enzyme, and 10-20 ng of gDNA. Genomic DNA was purified using either a Quick-DNA Miniprep Plus Kit (Zymo Research, Tustin, CA, cat# D4068) or by phenol:chloroform extraction and ethanol precipitation. The extraction method varied based on the quantity of the initial sample. Primer and probe sequences used for dPCR drop-off assays are listed in Supplementary Data 1. Quantstudio Absolute Q Software (v6.2) uses a Poisson distribution that allows for quantification of each amplified product in copies/µL based on the fluorescent signal in each microchamber. All statistical tests were determined using GraphPad Prism (v10.2.2).

### DNA Sequencing
For amplicon sequencing, (AmpSeq) DNA regions were amplified from Nalm6 and Reh gDNA using primers with partial Illumina adapter sequences (Supplementary Data 1). PCR reactions were performed using Platinum Taq DNA Polymerase (Invitrogen, Carlsbad, CA). Amplified DNA fragments were purified using HighPrep PCR clean-up magnetic beads (MAGBIO, Gaithersburg, MA, cat# AC-60250) at 1.8x concentration. SNP/INDEL detection analysis was performed by Genewiz's Amplicon E-Z Next Generation Sequencing (Azenta Life Science, South Plainfield, NJ). Genomic DNA was submitted to Novogene (Sacramento, CA) for long-read whole genome sequencing (WGS). Libraries were prepared using the PacBio Sequel II DNA HiFi kit and run on the PacBio Revio platform. Analysis of WGS was performed by Novogene. Briefly, raw reads were aligned to the reference sequence using PBMM2 (v1.8.0) that aligns PacBio data, outputs PacBio BAM files, and is a single molecule real-time sequencing (SMRT) minimap2 wrapper for PacBio data. Structural variant (SV) calling was performed using cuteSV[61] (v2.1.1) that detects all types of SVs using evidence from split-read alignments, high-mismatch regions, and coverage analysis. Annotation of variants was done using ANNOVAR[62]. Visualization of reads was done using Integrated Genomics Viewer (v2.17.2.01)

### High-Throughput Rejoin and Genome-Wide Translocation Sequencing (HTGTS-JoinT-seq)
HTGTS was performed using the linear amplification-mediated (LAM) platform[63] with modifications as described[63] to quantify both single DSB rejoining as well as translocation to other DSBs (JoinT-seq). Briefly, 10 µg of isolated gDNA was used to prepare amplicons for Illumina Nova-seq 150 bp paired end sequencing. Sequence reads were aligned to the hg38 genome build and normalized to 2,914,360 and 793,219 each for IGHM-6 and IGHM-1 bait analyses, respectively. Hotspot determination used MACS2 with an FDR-adjusted P-value cutoff of 10$^{-9}$ as described[64].

## ADMIXTURE Allele Frequency Projection Analysis

Infinium Global Diversity Array iScan data was converted from iDat to VCF then to PLINK ped/map format. dSNP annotation was performed to provide the proper RSID, as well as pruning for linkage equilibrium. A reference allele frequency map of 1000 Genome samples for each of the sub populations included in the study was constructed prior to running the admixture allele frequency projection onto the reference map to determine the admixture probability for each subpopulation[65].

## RNA isolation and qPCR

Total RNA was extracted from cells using TRIzol (Thermo Fisher Scientific, cat# 15596018) as described by the manufacturer's instructions. cDNA was generated using the Maxima H Minus cDNA Synthesis Master Mix (Thermo Fisher Scientific, cat# M1661) using 1 μg of total RNA according to the manufacturer's instructions. The cDNA was then used as a template in quantitative real-time PCR using a SYBR Green Master Mix (Applied Biosystems, cat# 100029283). Gene expression was normalized to 18 S ribosomal RNA. Primer sequences used for gene expression analysis are listed in Supplementary Data 1.

## RNA-seq Analysis

Poly-A enriched RNA was isolated from patient samples stored at −80 °C in TRIzol for library preparation and 150 bp paired-end sequencing was performed by Novogene (Beijing, China). FASTQ files for RNA-seq have been deposited in Sequence Read Archive (SRA, Bioproject PRJNA31257). FASTQ files were inspected for base quality scores using FASTQC (v0.11.9) and aligned to the human genome (GRCh38) using STAR aligner[66] (v2.7.10). From aligned BAM files, PCR duplicates were removed using MarkDuplicates in GATK[67] (v4.5.0.0). Next, counts matrices were assembled from BAM files in using the summarizeOverlaps function from the GenomicAlignments package[68]. Counts matrices were then filtered for genes with a rowSum of 10 counts across all samples then differential expression was performed using DESeq2[69] (v1.44.0). Differential expression analyses were performed by applying a logistic regression of effect on gene expression counts as a difference between Ph-like and Ph+ groups. These analyses used standard pipelines from the R package DESeq2 where significance is calculated using a Wald t-test and resulting distributions of p-values are adjusted for a false discovery rate of 1%.

## Western blotting

Protein lysates containing an equal amount of protein were loaded and resolved on an SDS−PAGE gel and transferred to a nitrocellulose membrane. For imaging, membranes were treated with Immobilon Western Chemiluminescent Substrate (Millipore, cat# WBKLS0500). Chemiluminescent images were obtained using the BioRad ChemiDoc System. Antibodies used for the Western blot were: AID (L7E7) (1:1,000 dilution) (Cell Signaling, cat# 4975, lot#1), FLAG M2 (1:1,000 dilution) (Sigma-Aldrich, cat# A2220, lot# 0000261540), and p84 (1:2,000 dilution) (GeneTex, cat# GTX70220, lot# 43948). Uncropped and unprocessed scans of western blots are provided in the Source Data file.

## Reporting summary

Further information on research design is available in the Nature Portfolio Reporting Summary linked to this article.

## Data availability

HTGTS data were deposited into the NCBO's Gene Expression Omnibus database under accession number GEO Series accession number GSE243667. RNA-seq data has been deposited in Sequence Read Archive (SRA) with the Bioproject accession number PRJNA31257 [https://www.ncbi.nlm.nih.gov/bioproject/?term=PRJNA1017867]. Long-read whole genome sequencing data are not publicly available due to their containing information that could compromise the privacy of research participants. Data availability is restricted to that which supports the findings of this study will be made available upon request to the corresponding author within one to six months. Source data are provided with this paper.

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

## Acknowledgements

This work was supported by grants from the National Cancer Institute (R37CA266042 and R01CA276470), American Cancer Society IRG-16-187-13, and the UCI Chao Family Comprehensive Cancer Center Anti-Cancer Challenge to N.R.P., and a National Cancer Institute Diversity Supplement to V.R. Financial support for the Masri lab is provided by NIH grants R01CA244519 and R01CA259370. Financial support for the Seldin lab is provided by NIH grants DP1DK130640 and R21AR082842. The authors are grateful to the patient donors and healthy volunteers that allowed their specimens to be part of this study.

## Author contributions

V.R. and J.N.S performed research, designed experiments, analyzed data, and contributed to writing the paper. A.G., V.M., M.L.F, S.E.C., Y.E.G., J.W., and E.M.S performed experiments and analyzed data. O.S.E and J.B.V. contributed to accessing patient material and information. S.P.T. provided epidemiological data on patient cohorts and catchment area. F.Q.R. supplied pathology information about rearrangements in patient samples and contributed to paper editing. S.M. contributed to qPCR and RNA experiments and editing of the paper. M.M.S. performed analysis on RNA-seq data. R.L.F. oversaw and analyzed HTGTS data. A.G.F provides patient material through her heme-biorepository and is PI on the IRB protocols. N.R.P conceived and directed the project, oversaw the results, and wrote the paper.

## Competing interests

The authors declare no competing interests.
