## [Peer Review File · Nature Communications]

Increased AID Results in Mutations at the CRLF2 Locus
Implicated in Latin American ALL Health DisparitiesREVIEWER COMMENTS

Reviewer #1 (Remarks to the Author):

AID plays a role in the induction of double strand breaks in B cell malignancies. Previous reports have indicated that Latin American ALL patients (LAs) more frequently show a CRLF2::IGH fusion while Europeans and non-LAs more often have an interstitial deletion between CRLF2 and P2RY8. The authors delineated a genome region 16 kb upstream of CRLF2 as a recombination hot spot for AID-induced double strand breaks. They used a doxycycline inducible AID expression cassette via lentiviral transduction to increase AID expression in NALM6 cells and found increased indels in this hot spot region, as revealed by digital PCR.

The authors compared amplicon sequencing with digital PCR for detecting AID induced indels and found a clear advantage to using the latter method, also because copy number variations are more difficult to detect by amplicon sequencing. The authors suggest that the reported higher prevalence of Ph-like ALL with CRLF2::IGH fusion is due to increased AID activity. In addition, they observed increased indels in an AID break cluster (ABC) near the BCL2 locus in non-LA patients.

The work is of interest and the methodology appears adequate.

The following points should be addressed:

1. What was the basis for the diagnosis of Ph-like ALL in the investigated patient cases (please include information in Table 1)? The authors should also provide a more detailed immunophenotype.

2. Line 333–335: "Overall, this indicates that LA patients presenting with increased instability at ABC sites are at higher risk for B cell cancers, ..." – this wording is not optimal. The authors have investigated patients with B cell cancers and detected a higher rate of instability at ABC sites but it is not clear if this instability was present before the onset of disease.

3. If increased instability at ABC sites / higher AID levels are the reason for the higher prevalence of Ph-like ALL in LAs, as the authors seem to suggest: do the authors have any idea about the cause of increased AID activity? Why isn't a similarly higher prevalence of follicular lymphoma observed in Latin American patients?

3. The authors should clearly indicate the identified ABC sites near CRLF2 and BCL2 in genomic coordinates.

4. Line 405–406: "... it is possible that further testing of patient 19-035 would confirm a Ph-like ALL diagnosis." – this is a speculation.

5. Line 414–415: "... likely indicate why Ph-like ALL (at least 2/3 of patients) is a much more aggressive disease than Ph+ ALL." – The generalized statement that "Ph-like ALL is a much more aggressive disease than Ph+ ALL" is not true. One major difference between Ph+ ALL and Ph-like ALL is the fact that there exists an effective targeted therapy for Ph+ ALL (ABL1 tyrosine kinase inhibitors) which is not the case in many cases of Ph-like ALL. But those cases of Ph-like ALL, where tyrosine kinases such as PDGFRB, ABL2, NTRK3, etc. are involved, can also very effectively be treated with TKIs. Ph-like ALL also certainly does not comprise "at least 2/3 of patients".

6. Line 396–398 "This further emphasizes that confirmation of the Philadelphia chromosome should still be followed-up by testing for CRLF2 overexpression or CRLF2 rearrangements since standard therapy for Ph+ ALL may be less effective in this situation." – this is speculative. Do the authors have data showing CRLF2::IGH also in Ph+ ALL patients?

7. Line 453–455: "Furthermore, it would provide a more rapid indicator over current standards to

allow patients to be more quickly put on an effective treatment regimen." – The authors should explain what they mean in more detail.

8. Line 460-463: "Currently several different assays need to be conducted to confirm a Ph-like ALL diagnosis including chromosomal microarray, next generation sequencing (NGS), flow cytometry increased CRLF2 receptor at the cell surface, and FISH analysis of cytogenetically cryptic rearrangements." – this is not entirely correct. The diagnosis of Ph-like ALL is currently mainly based on RNA-seq data revealing aberrantly expressed or activated kinases.

9. In Table 1 in patient 19-035 genetics/FISH was "not available". How could the authors be sure that the patient was Ph-negative?

10. One major point of critique is the number of investigated patients (effectively two Ph-like patients).

General remarks:

- the authors should use modern gene terminology (e.g., in Table 1), and the double colon notation for fusion genes: BCR::ABL1 instead of BCR-ABL, KMT2A::AFF1 instead of MLL-AF4, etc.
- Some minor typos: line 389 "associate with ALL", line 400 "preformed"

Reviewer #2 (Remarks to the Author):

In this manuscript, Rangel et al have developed a digital PCR (dPCR) assay to track AID-induced lesions at two loci-CRLF2 and Bcl2-that are known to be targeted by AID in the context of B cell ALL. Using this technique, the authors were able to detect AID signatures in ALL patients. The authors posit that dPCR can be a convenient tool to detect AID-induced mutations in ALL patients and could contribute to the early detection of the disease, which is disproportionately affects the Latin American population.

AID-dependent lesions have been strongly associated with mutations and translocations associated with many types of B cell malignancies and the idea that (inadvertent or physiological) AID expression in developing B cells cooperate with RAG-induced lesions has been proposed for years; however, there is not enough data to support this idea. Thus, a robust method to detect mutations at the CRLF2 and Bcl2 regions is of much relevance. However, the authors should address the following issues to increase the impact of the work.

Fig. 1. The authors note that AID expression in the Nalm6 cells is below the level of detection, yet they find signatures of AID activity. It is unclear if these truly represent AID activity at these regions or the activity of some other APOBECs or just replication-induced lesions. The authors should attempt to delete AID from the Nalm6 or Reh lines they have used to examine if the mutations detected are truly AID-dependent.

Fig. 3. Here the authors try to correlate AID-induced mutations at CRLF2 and Bcl-2 in Nalm6 cells over-expressing AID from a lentiviral vector. This is one of the stronger figures of the manuscript. However, it is unclear how specific AID-induced mutations are for the two test regions, given that AID has been shown to mutate many sequences beyond the Ig loci. The authors used a couple of control regions but it is unclear how much power they have in this rather restricted analysis to conclude that CRLF2 and Bcl-2 are specific AID targets that can be detected by dPCR.

Fig 4 and 5 employ dPCR to examine the mutation profile of the target genes. The authors clearly can use the technique to pick up mutations, but again the results are only correlative as there is no data

on AID expression. Unless there is a way to detect AID expression, even by qPCR, it is difficult to interpret the data.

Fig. 6 compares the transcriptome of Ph-ALL patient cohorts. It is unclear what the data suggests. There are bound to be transcriptomic differences between different patients. And again, in the absence of any data suggesting AID is expressed in these leukemias, it is difficult to interpret the data.

Reviewer #3 (Remarks to the Author):

The manuscript by Ranger and Sterrenberg et al establishes a new digital PCR (dPCR) assay to detect the mutational landscape characterising acute lymphoblastic leukaemia (ALL). The authors focus their study on a subtype of ALL that shows increased prevalence in Latin Americans and lacks BCR-ABL1 chromosomal translocations typical of Philadelphia-positive ALL; yet shares similar patterns of gene expression with Ph positive ALL, therefore referred to as Ph-like ALL. An established dPCR-based assay would constitute a rapid and easy-to-access diagnostics method for Ph-like ALL, possibly improving early detection and treatment of this heterogeneous group of ALL patients for which standard treatment is often a poor choice. Similarly, a dPCR carries the potential to shed light into the aetiology of Ph-like ALL. Evidence suggests a role for AID-induced mutation in bone marrow B-cell precursors, and dPCR allows a more detailed analysis of the regions where AID-induced double-strand DNA breaks are known to occur.

The authors initially use cell lines to characterise regions of high AID-induced mutational burden - AID break clusters (ABCs) - in CRLF2 and BCL2 loci, comparing amplicon sequencing to dPCR. Overexpression of AID is used to determine whether AID activity correlates with increased genomic instability at CpG sites on CRLF2 and BCL2 ABCs. A few questions arise from these analyses:

1. Do the levels of control probe SUN stay constant with AID overexpression? I note that dPCR in CRLF2 ABC shows that TAMRA and Cy5 probes have increased drop-off with AID overexpression and SUN probe is located between the two. Depending on the DNA rearrangements at play, there may be a decrease in the signal for SUN.
2. dPCR is based on fluorescent signal in individual drops/DNA molecules. How do the various fluorescent probe signals compare to each other? Do the authors find evidence for particular combinations of probe drop-offs, i.e. do they happen on the same molecule? There is a greater level of detail here that is not explored/showed and could be informative.
3. Mapping of AmpSeq primers for the BCL2 ABC (Supplementary Figure 1B) would be helpful; the point about the SUN probe made earlier also applies here, particularly with amplicon sequencing showing rearrangement downstream of the 3rd CpG site/FAM probe.
4. It's unclear how the LAM-HTGTS results constitute a significant addition to the manuscript; similar analyses in Nalm6 cells with AID overexpression could inform whether increased genomic instability brought about by AID-induced mutation results in chromosomal translocations.

When analysing patient samples, the authors provide a compelling case for the outperformance of dPCR over amplicon sequencing, but only two Ph-like ALL samples are used.

1. It is unclear how predictive value of dPCR analyses in the CRLF2 ABC regions for Ph-like ALL - i.e. drop-off rates can vary within patients of the same group, and yet show similar patterns between patients of different groups (for eg, patients 17-062 (Ph-like) and 15-010 (Ph positive) show very similar drop-off levels in Figures 4D and E). While it is understandable that Ph-like ALL samples are rare the small-scale study carried out by the authors doesn't provide evidence that dPCR can discriminate between Ph-like and Ph positive ALL samples.
2. Although figures 4 and 5 refer to patients 17-062 and 19-021 as Ph negative, I assume these are Ph-like as stated in Table 1. Do Ph negative patients (eg 17-004, which doesn't show increased

expression of CRLF2) also show genomic instability at the CRLF2 locus?

3. The authors state in the manuscript 'dPCR could provide an early indicator of disease' but the analyses shown are performed in samples with an overload of B-ALL cells. There is no indication dPCR will have the sensitivity to detect small numbers of pre-ALL cells in individuals at risk of developing disease. This is an important consideration if dPCR were to be adopted for early detection.

Overall, I find the study useful in showing dPCR is a powerful methodology to characterise genomic instability in ALL samples. But it doesn't provide compelling evidence supporting the use of dPCR to discriminate between different ALL subtypes or detect pre-disease states and these are key requirements for dPCR to be adopted as a new diagnostic tool. Likewise, other than correlating AID overexpression with increased genomic instability at CRLF2 and BCL2 loci, this study doesn't significantly advance our understanding of the aetiology of ALL.

Point-by-Point Response to Reviewer Comments.

Rangel and Sterrenberg, et. al, *Increased AID Results in Mutations at the CRLF2 Locus Implicated in Latin American ALL Health Disparities*

We thank each of the reviewers for their time in providing insightful and constructive feedback on our study. Very good points were raised by all three reviewers, and we have put in tremendous effort to address each of these comments with the overall result being a much stronger study. Briefly, a major issue raised was the limited number of Ph-like ALL patients in our study. By further utilizing our UCI biobank and consenting additional patients, we have added 8 additional Hispanic Ph-like ALL samples, 4 with the *CRLF2::IGH* translocation and 4 with the *P2RY8::CRLF2* intra-chromosomal deletion. This has vastly improved the comparative power of our study between Ph-like and Ph+ ALL. The patient analysis shown in Figure 5 is completely new to this revision, as are several panels in Figure 3 and 4. We have also provided additional supporting evidence in our supplementary data. Altogether, we provide stronger evidence of AID activity at *CRLF2* and greater instability at *CRLF2* in Hispanic patients with a *CRLF2::IGH* translocation versus those with the *P2RY8::CRLF2* rearrangement or with Ph+ disease. Our additional data also suggests *CRLF2* is a region prone to instability in normal B cell development. Below is our detailed point-by-point response with original reviewer comments in **black** and our response in **blue**. We have also added new data where appropriate to facilitate evaluation of our responses.

Reviewer #1 (Remarks to the Author):

AID plays a role in the induction of double strand breaks in B cell malignancies. Previous reports have indicated that Latin American ALL patients (LAs) more frequently show a *CRLF2::IGH* fusion while Europeans and non-LAs more often have an interstitial deletion between *CRLF2* and *P2RY8*. The authors delineated a genome region 16 kb upstream of *CRLF2* as a recombination hot spot for AID-induced double strand breaks. They used a doxycycline inducible AID expression cassette via lentiviral transduction to increase AID expression in NALM6 cells and found increased indels in this hot spot region, as revealed by digital PCR. The authors compared amplicon sequencing with digital PCR for detecting AID induced indels and found a clear advantage to using the latter method, also because copy number variations are more difficult to detect by amplicon sequencing. The authors suggest that the reported higher prevalence of Ph-like ALL with *CRLF2::IGH* fusion is due to increased AID activity. In addition, they observed increased indels in an AID break cluster (ABC) near the *BCL2* locus in non-LA patients.

The work is of interest and the methodology appears adequate.

We thank the reviewer for their supportive and thoughtful comments.

The following points should be addressed:

1. What was the basis for the diagnosis of Ph-like ALL in the investigated patient cases (please include information in Table 1)? The authors should also provide a more detailed immunophenotype.

We have improved **Table 1** by adding additional information on how the Ph-like ALL diagnosis was made. Our major focus is on Ph-like ALL involving rearrangements at *CRLF2*, thus flow cytometry was used to detect increased *CRLF2* expression as measured by increased receptor

on the cell surface. FISH was performed using break apart probes to detect rearrangements at the *CRLF2* and *IGH* loci and the *CRLF2* and *P2RY8* loci. The patient coded as Ph-like-7 had RNA sequencing done to confirm a *P2RY8::CRLF2* fusion transcript. Importantly, final determination that patient Ph-like-4 had a *CRLF2::IGH* translocation came from this study and the long-read sequencing done on this patient (Fig. 5C)

Also, in addressing reviewer's comment #10, we reformatted Table 1 as we have been able to add 8 more Ph-like ALL patients to our study that we have collected since the first submission. This includes 2 additional Hispanic patients with *CRLF2* and *IGH* rearrangements. 4 Hispanic patients with *P2RY8* and *CRLF2* rearrangements and a non-Hispanic Asian and a non-Hispanic White patient with *CRLF2* and *IGH* rearrangements. To try and make the patient coding easier to follow, they are now identified as Ph-like-X, Ph+-X, or ALL-X based on their ALL subtype. So that the reviewers can distinguish the newly added patients from the patients that were in the 1st submission we also include a key that shows the original name from the 1st submission. We have also included additional healthy White and Hispanic donor as well (H-X-X). The samples from the 1st submission are highlighted in yellow in the Patient Key below.

Table 1. De-Identified Material Collected from ALL Patients Treated at the University of California, Irvine

Sample	Age	Sex	Ethnicity	Genetics/FISH Findings	Source	% Blasts
Ph-Like ALL¹						
Ph-like-1	60	M	Hispanic	CRLF2 and IGH Rearrangement ²	BM	95
Ph-like-2	24	M	Hispanic	CRLF2 and IGH Rearrangement ²	BM	95
Ph-like-3	70	F	Hispanic	CRLF2 and IGH Rearrangement ²	BM	95
Ph-like-4	54	M	Hispanic	CRLF2::IGH ³	PB	90
Ph-like-5	31	F	Hispanic	P2RY8 and CRLF2 Rearrangement ²	BM	62
Ph-like-6	46	F	Hispanic	P2RY8 and CRLF2 Rearrangement ²	BM	50
Ph-like-7	70	M	Hispanic	P2RY8::CRLF2 ⁴	PB	59
Ph-like-8	26	M	Hispanic	P2RY8 and CRLF2 Rearrangement ²	PB	90
Ph-like-9	29	M	Asian	CRLF2 and IGH Rearrangement ²	BM	56
Ph-like-10	75	M	White	CRLF2 and IGH Rearrangement ²	PB	91
Ph-Positive ALL						
Ph+-1	64	M	Hispanic	BCR::ABL1	BM	95
Ph+-2	44	M	Hispanic	BCR::ABL1	BM	90
Ph+-3	78	F	Asian	BCR::ABL1	BM	90
Ph+-4	50	F	Asian	BCR::ABL1	BM	90
Ph-Negative ALL						
ALL-1	48	M	Hispanic	Negative for BCR::ABL1 , IGH , CRLF2 , MYC , and KMT2A Rearrangements	PB	73
ALL-2	61	M	Hispanic	KMT2A::AFF1	BM	80

¹Ph-like ALL was diagnosed using flow cytometry to detect increased *CRLF2* expression.

²Break-apart probes were used to detect *CRLF2* and *IGH* rearrangements

³Confirmed by long-read sequencing in this work (see Fig. 5C)

⁴Next generation sequencing was used to detect a *P2RY8::CRLF2* fusion RNA transcript

Patient Key (reviewers only)

Sample	NEW	Ph	Race/Eth
17-062	Ph-like-1	Ph-like Ig	His
19-021	Ph-like-2	Ph-like Ig	His
15-010	Ph+-1	Ph+	His
17-061	Ph+-2	Ph+	His
19-035	ALL-1	Ph-	His
17-004	ALL-2	Ph-	His
17-020	Ph+-3	Ph+	Asian
16-022	Ph+-4	Ph+	Asian
UCI-001	Ph-like-5	Ph-like P2	His
UCI-002	Ph-like-6	Ph-like P2	His
UCI-004	Ph-like-9	Ph-like Ig	Asian
UCI-005	Ph-like-10	Ph-like Ig	Non-His
UCI-006	Ph-like-3	Ph-like Ig	His
NP-001	Ph-like-7	Ph-like P2	His
NP-004	Ph-like-4	Ph-like Ig	His
NP-005	Ph-like-8	Ph-like P2	His
p-001	H-W-4	N/A	White
p-002	H-H/L-1	N/A	His
p-003	H-H/L-2	N/A	His
p-004	H-H/L-3	N/A	His
p-005	H-W-1	N/A	White
p-006	H-A-1	N/A	Asian
p-007	H-A-2	N/A	Asian
p-008	H-H/L-4	N/A	His
p-009	H-W-2	N/A	White
p-010	H-W-3	N/A	White
p-011	H-H/L-5	N/A	His

Ph-Like-4

Fig. 5C showing t(X;14)

2. Line 333–335: “Overall, this indicates that LA patients presenting with increased instability at ABC sites are at higher risk for B cell cancers, ...” – this wording is not optimal. The authors have investigated patients with B cell cancers and detected a higher rate of instability at ABC sites but it is not clear if this instability was present before the onset of disease.

We agree with the reviewer that this statement may not be the most accurate, especially in light of new data added to this revision. This section has been extensively revised, and this line no longer appears. Our original intention was to correlate that inducing AID expression in our cell lines increases mutations at the *CRLF2* and *BCL2* ABC sites (**Fig. 3**) with the instability present in the patient samples. The reviewer is correct, though, that this is based on patients that already have cancer. That we have added to the number of Ph-like ALL patients and healthy donors in this revision has increased our clarity and allowed for more specific comparisons. We now directly compare Ph-like ALL patients with either a *CRLF2::IGH* translocation or a *P2RY8::CRLF2* deletion in new **Fig. 5A**. Previous studies (PMID: 20847213) have shown that the *CRLF2-IGH* translocation is likely AID driven while the *P2RY8::CRLF2* deletion is not and here we clearly see increased drop off at the AID target regions of *CRLF2* in the patients with the translocation, but not the intrachromosomal deletion.

To speak directly to the reviewer’s point about whether instability is present prior to the onset of disease, we now perform dPCR at the *CRLF2* and *BCL2* loci in 5 healthy Hispanic and 6 healthy non-Hispanic patients. As we were able to obtain larger volumes of fresh DNA from these donors, we were able to separate the CD19+ B cell population from the CD19- population. Strikingly, gDNA from the CD19+ cells have a higher level of mutations at *CRLF2* vs the CD19- population and this is only significant in the Hispanic population. This is not seen at *BCL2* as the level of mutations in CD19- vs CD19+ cells are comparable except at one AID-target site in Hispanics. These data (**Figs. 5E and 5F below**) show that *CRLF2* is vulnerable to AID in developing B cells and this may be a driver of instability and *CRLF2::IGH* formation.

Fig. 5A

Fig. 5E

Fig. 5F

3. If increased instability at ABC sites / higher AID levels are the reason for the higher prevalence of Ph-like ALL in LAs, as the authors seem to suggest: do the authors have any idea about the cause of increased AID activity? Why isn't a similarly higher prevalence of follicular lymphoma observed in Latin American patients?

This is indeed a fascinating aspect of this project. Our major purpose for this study was to demonstrate that AID can target the ABC sites when it is present in B cells and to show that our novel dPCR assay can detect these changes, which we feel we have accomplished based on our revised Fig. 3 using a pre-B cell model. When AID goes from undetectable to high expression, we see clear changes at AID-sensitive sites at both *CRLF2* and *BCL2*, but interestingly, not at all of the AID-sensitive sites, suggesting that something beyond just sequencing is dictating AID activity.

Figure 3: Rangel, et al

The next step was to see if this was recapitulated in patient samples, which we surprisingly found was not. When we examine patient samples and compared the *CRLF2::IGH* vs *P2RY8::CRLF2*, the *CRLF2::IGH* patients have significantly higher instability at *CRLF2*, suggesting these samples have higher AID activity since the AID-sensitive sites are being mutated. If we examine the *BCL2* locus in the same patients, however, we do not see increased drop-off.

Fig. 5A. Drop-off at *CRLF2*

Fig. S7. Drop-off at *BCL2*

Similarly, when we examined the CD19- vs CD19+ populations from healthy donors we found that Hispanics had significantly higher drop-off at *CRLF2* in CD19+ cells, but also that non-Hispanics had elevated, though not significant drop-off at *CRLF2* as well. In contrast, the *BCL2* locus did not show this increased drop off.

Fig. 5E and 5F

What these results and other recent studies suggest is that factors beyond just DNA sequence are dictating sensitivity to AID and most likely DNA methylation is a factor (PMID: 25624348, 34260910). AID can deaminate both methylated and unmethylated cytosines. While AID converts C→U, it converts meC→T with these T's being repaired more slowly leading to a longer-lived lesion (PMID: 29735400, 38490833). While we have preliminary data supporting this, the role of DNA methylation in Ph-like ALL disparities requires more extensive studies and will be a focus of our future work.

Based on the reviewer's comments, we have also added this paragraph to the discussion:

Interestingly, the *BCL2* locus was not targeted in either the Ph-like ALL patients or the healthy donors. (Fig. 5F, Supplementary Fig. 7). On the one hand, this makes sense as follicular lymphoma is not associated with a Hispanic health disparity, but it raises a critical question as to why *BCL2* can be targeted in the Nalm6 cell line, but not in the patient samples. A strong possibility is that factors beyond just the primary DNA sequence dictate sensitivity to AID. This has already been demonstrated by the dPCR performed in Nalm6 cells as not all AID target sequences were equally hit (Fig. 3). DNA methylation can affect the accessibility and vulnerability of DNA sequences to AID and previous work has shown variable or partial methylation in ABC sites including *CRLF2* and *BCL2*^{9,10,65,66}. Thus, interrogating the methylation status of the *CRLF2* and other ABCs will be critical to fully understanding how AID can target regions and lead to oncogenic translocations.

3. The authors should clearly indicate the identified ABC sites near CRLF2 and BCL2 in genomic coordinates.

We agree with the reviewer that this is important information. We now indicate the chromosomal coordinates for the *CRLF2* and *BCL2* ABCs both in the main text and in Supplementary Figure 1A:

*The detailed sequence for the CRLF2 region with the ABC (ChrX: 1,228,310-1,228,969, hg38) is shown in **Supplementary Fig. 1A**.*

*Similarly, using AmpSeq at the BCL2 ABC (Chr18: 63,126,040-63,126,870, hg38) (**Supplementary Fig. 1B**), we also see indels, yet observe a notably different pattern for each cell line (**Fig. 1C**).*

Supplementary Figure 1. DNA sequences of the ABCs at CRLF2 and BCL2 (A) *CRLF2 ABC sequence, Chr X: 1,228,310-1,228,969 (same coordinates for Chr Y). Orange arrows indicate sequences of amplicon primers for AmpSeq and yellow arrows are amplicon primers for dPCR. Blue, yellow, green, and red boxes indicate sequences where the FAM, TAMRA, SUN, and Cy5 drop-off probes bind, respectively. Vertical arrows indicate cut sites targeted by Cas9 via sgCRLF2-1 and sgCRLF2-3. (B) BCL2 ABC sequences, Chr 18: 63,126,040-63,126,870. Orange arrows indicate sequences of amplicon primers for AmpSeq and yellow arrows indicate sequences of amplicon primers for dPCR. Red, blue, and green boxes indicate sequences where the Cy5, FAM, and HEX drop-off probes bind, respectively. For simplicity, only one strand is shown, so the reverse amplicon primers are the reverse complement. For all exact sequences, see Supplemental Table 1. All chromosome coordinates are from the December 2013 GRCh38/hg38 sequencing build. dPCR program: 96°C for 10 minutes (1x); 96°C for 10 seconds, 63°C for 20 seconds, 68°C for 30 seconds (40x).*

4. Line 405–406: "... it is possible that further testing of patient 19-035 would confirm a Ph-like ALL diagnosis." – this is a speculation.

We agree with the reviewer that it is important to provide additional evidence to support this. While the *CRLF2* expression in patient ALL-1 (referred to as 19-035 in the previous submission) is comparable to what is measured in the Ph-like ALL patients (**Fig. 4A**), our clinical collaborators were able to determine from additional clinical diagnostics in this patient's chart that they were negative for a *CRLF2* rearrangement (**Table 1**). Thus, we have removed this statement as we do not have definitive evidence to support that this patient, now ALL-1, has a *CRLF2* rearrangement. In the future, we hope to perform more long-read sequencing of material as resources permit to fully map all rearrangements in patients. While we note that ALL-1 matches the expression signatures seen in the Ph-like and not the Ph+ cases, we avoid any additional speculation on this patient.

5. Line 414–415: "... likely indicate why Ph-like ALL (at least 2/3 of patients) is a much more aggressive disease than Ph+ ALL." – The generalized statement that "Ph-like ALL is a much more aggressive disease than Ph+ ALL" is not true. One major difference between Ph+ ALL and Ph-like ALL is the fact that there exists an effective targeted therapy for Ph+ ALL (ABL1 tyrosine kinase inhibitors) which is not the case in many cases of Ph-like ALL. But those cases of Ph-like ALL, where tyrosine kinases such as PDGFRB, ABL2, NTRK3, etc. are involved, can also very effectively be treated with TKIs. Ph-like ALL also certainly does not comprise "at least 2/3 of patients".

We appreciate the reviewer’s feedback in making this point more accurate. We have removed this statement and instead state:

Several of the most significant pathways affected in Ph-like patients include those that regulate kinase activation, which aligns with previous reports that distinguish CRLF2-linked ALL from Ph+ ALL. Utilization of multiple kinase pathways may make Ph-like ALL more refractory to treatments that only target a single kinase pathway⁵⁵⁻⁵⁷.

Thus, instead of using the vague term “aggressive” we reference data showing that targeting the CRLF2 driven kinase pathway alone with JAK inhibitors has been shown to be ineffective in some cases of Ph-like ALL without treatment using additional TKIs (PMIDs: 32191635, 37124486, 29487712), which fits with our pathway enrichment analysis in **Fig. 4C**. Relevant as well is that Hispanics and non-Hispanics do not respond to treatment regimens equally (PMID: 37933194), adding an additional layer to Ph-like ALL treatment outcomes.

Fig. 4C

Also, for clarification, we were not suggesting the Ph-like ALL comprises 2/3 of ALL. We were stating that 2 out of 3 of the non-Ph+ samples we submitted for RNA-seq were Ph-like ALL cases with CRLF2 and IGH rearrangements. We removed this to reduce any confusion as our major DEG comparison is only between the Ph-like and Ph+ patients and apologize for the poor phrasing.

6. Line 396–398 "This further emphasizes that confirmation of the Philadelphia chromosome should still be followed-up by testing for CRLF2 overexpression or CRLF2 rearrangements since standard therapy for Ph+ ALL may be less effective in this situation." – this is speculative. Do the authors have data showing CRLF2::IGH also in Ph+ ALL patients?

The potentially serious nature of having *CRLF2::IGH* in Ph+ patients was recently demonstrated by Jain, et al. (PMID: 28860345) and we feel it important to highlight this possibility. Furthermore, due to the greater number of patient samples we have, and the more thorough analysis, this section of the results has been extensively revised. We found that two patients, Ph-like-4 and Ph+2 had dPCR results that were drastically different from the other patients and support further instability of the *CRLF2* ABC. **Fig. 5B** shows that there is a substantial and significant increase in the SUN drop-off product (*i.e.*, amplicons that can no longer bind the SUN probe). As we state in the text, we might expect this level of instability for a Ph-like ALL patient with a *CRLF2::IGH* product. Even with the translocation being reciprocal, loss of some genetic material might be possible near a translocation junction. For the Ph+2 patient, however, we would not expect *CRLF2* to be affected. Also, Ph+2 also had high expression of *CRLF2* (**Fig. 4A**) and *AICDA* (**Fig. 4B**).

Fig. 5B

Fig. 4A

Fig. 4B

In our first submission, we highlighted Ph+-2 (formally 17-061) as an example of how amplicon sequencing is not as sensitive as dPCR for detecting changes. For most of the amplicon sequencing done with patient samples, the indel pattern does not change, even though for Ph+-2, we can see through dPCR that the total SUN signal drops by approximately half compared to the FAM, TAMRA, and CY5 signals, hence the high level of drop-off product (Former Supplementary Fig. 4 below). This is why we did not perform amplicon sequencing on the patient samples added to this revision.

reviewers
only

Former Supplementary Figure 4. Total fluorescent signal from each probe from dPCR of patient 17-061. (A) AmpSeq data indicating indels from patient 17-061. (B) Quantification of total fluorescent signal from each probe (i.e., not drop-off product as indicated in histograms elsewhere, but total fluorescence level). Indicates mean values \pm SEM from at least 6 technical replicates. (C) Raw fluorescent data shown as 1D blots. Dots below the light blue threshold line indicate microchambers with no signal from the indicated fluorophore and dots above the threshold indicate a microchamber positive for that fluorescent signal with the amplitude of the signal indicated on the y-axis. These values were used to calculate the copies/ μ L values in B (one representative run from 6 replicates) where there is half as much SUN signal compared to the other 3 fluorescent signals.

Since, out of the dPCRs we performed, these were the only two patient samples with such a pattern, we decided to perform long-read whole genome sequencing through Novogene using their PacBio Revo system. Our analysis was limited to these two samples given how cost prohibitive this analysis is (\$3,000-\$4,000 per genome). For Ph-like-4, we were able to receive adequate read depth to be able to clearly map the *CRLF2::IGH* translocation as described above. Unfortunately, the DNA from patient Ph+-2 was not of good enough quality to provide adequate coverage. Since we used the last of this sample for the sequencing, we were unable to obtain more. What we can clearly see from **Fig. 5D** is that the Ph+-2 patient is not as unstable as Ph-like-4 and that the *BCR::ABL1* translocation is clearly present. When we compare the reads from each patient on IGV (**Supplementary Fig. 8**), there is a large deletion on Chr. 14 in Ph+-2 at the same location as the translocation junction in Ph-like-4 and this appears in **Fig. 5D**. The lack of read-depth at Chr X, though, may be a factor in not seeing any split reads, thus not allowing us to conclusively say if there is a translocation at this site.

Ph-Like-4

Fig. 5C

Ph+-2

Fig. 5D

Chr 14: 105,846,756-105,973,317

Chr X: 1,180,861-1,273,107

Supplementary Fig. 8

To summarize, dPCR and qPCR data support that patient Ph+2 has irregularities in the *CRLF2* sequence and *CRLF2* and *AICDA* expression, but WGS was unable to definitively confirm a *CRLF2::IGH* translocation. There are clearly breakpoints present at *IGH*, but the read-depth is too shallow at *CRLF2*.

7. Line 453–455: "Furthermore, it would provide a more rapid indicator over current standards to allow patients to be more quickly put on an effective treatment regimen." – The authors should explain what they mean in more detail.

We wanted to convey the potential use of dPCR in detecting signatures linked to AID activity and have changed this to:

Future studies will further add to our cohort of Hispanic Ph-like ALL patients to determine if AID signatures are a consistent feature of cancers linked to ABC sites. COSMIC signatures indicative of AID activity include SBS84 and SBS85,⁶⁵ but determining this signature in a patient requires whole genome sequencing with a matched normal sample, which is cost prohibitive. dPCR may offer a rapid and cost-effective alternative to measure instability at sites proposed to be involved in cancer etiology that are sensitive to aberrant AID activity.

8. Line 460-463: "Currently several different assays need to be conducted to confirm a Ph-like ALL diagnosis including chromosomal microarray, next generation sequencing (NGS), flow cytometry increased *CRLF2* receptor at the cell surface, and FISH analysis of cytogenetically

cryptic rearrangements." – this is not entirely correct. The diagnosis of Ph-like ALL is currently mainly based on RNA-seq data revealing aberrantly expressed or activated kinases.

We thank the reviewer for this clarification. To make this statement more accurate, we have changed it to:

Currently several different assays can be conducted to confirm a Ph-like ALL diagnosis. Following an initial test using RNA-seq for detection of fusion transcripts and aberrantly expressed kinases, additional methods can be used to determine specific translocations and mutations including flow cytometry for increased CRLF2 receptor at the cell surface and FISH analysis of cytogenetically cryptic rearrangements⁶⁵.

9. In Table 1 in patient 19-035 genetics/FISH was "not available". How could the authors be sure that the patient was Ph-negative?

We agree that this seems ambiguous. We used "not available" meaning that the rearrangement present is unknown, though testing showed it to be Ph-negative. The original pathology report shows that FISH was performed and found to be negative for *BCR::ABL1* using a tri-color, dual-fusion probe. Using break apart probes, the patient was also found to be negative for *IGH*, *CRLF2*, *KMT2A*, and *MYC* rearrangements. The report goes on to state, "These studies **did not detect** aberrations in the 200 nuclei/probe examined" in the ALL panel. We have updated Table 1 to reflect this and be more specific.

10. One major point of critique is the number of investigated patients (effectively two Ph-like patients).

We recognize that this was a critical aspect of our study that needed to be addressed in our revision even given the limitation that the Ph-like ALL cases from which we can derive adequate samples can be rare. As detailed above, we have expanded the number of Ph-like ALL patients in our study from 2 to 10, which includes 4 Hispanics with *CRLF2* and *IGH* rearrangements and 4 Hispanics with *P2RY8* and *CRLF2* rearrangements:

Table 1. De-Identified Material Collected from ALL Patients Treated at the University of California, Irvine

Sample	Age	Sex	Ethnicity	Genetics/FISH Findings	Source	% Blasts
Ph-Like ALL¹						
Ph-like-1	60	M	Hispanic	CRLF2 and IGH Rearrangement ²	BM	95
Ph-like-2	24	M	Hispanic	CRLF2 and IGH Rearrangement ²	BM	95
Ph-like-3	70	F	Hispanic	CRLF2 and IGH Rearrangement ²	BM	95
Ph-like-4	54	M	Hispanic	CRLF2::IGH³	PB	90
Ph-like-5	31	F	Hispanic	P2RY8 and CRLF2 Rearrangement ²	BM	62
Ph-like-6	46	F	Hispanic	P2RY8 and CRLF2 Rearrangement ²	BM	50
Ph-like-7	70	M	Hispanic	P2RY8::CRLF2⁴	PB	59
Ph-like-8	26	M	Hispanic	P2RY8 and CRLF2 Rearrangement ²	PB	90
Ph-like-9	29	M	Asian	CRLF2 and IGH Rearrangement ²	BM	56
Ph-like-10	75	M	White	CRLF2 and IGH Rearrangement ²	PB	91
Ph-Positive ALL						
Ph+1	64	M	Hispanic	BCR::ABL1	BM	95
Ph+2	44	M	Hispanic	BCR::ABL1	BM	90
Ph+3	78	F	Asian	BCR::ABL1	BM	90
Ph+4	50	F	Asian	BCR::ABL1	BM	90
Ph-Negative ALL						
ALL-1	48	M	Hispanic	Negative for BCR::ABL1 , IGH , CRLF2 , MYC , and KMT2A Rearrangements	PB	73
ALL-2	61	M	Hispanic	KMT2A::AFF1	BM	80

¹Ph-like ALL was diagnosed using flow cytometry to detect increased *CRLF2* expression.

²Break-apart probes were used to detect *CRLF2* and *IGH* rearrangements

³Confirmed by long-read sequencing in this work (see Fig. 5C)

⁴Next generation sequencing was used to detect a *P2RY8::CRLF2* fusion RNA transcript

Patient Key (reviewers)

Sample	NEW	Ph	Race/Eth
17-062	Ph-like-1	Ph-like Ig	His
19-021	Ph-like-2	Ph-like Ig	His
15-010	Ph+1	Ph+	His
17-061	Ph+2	Ph+	His
19-035	ALL-1	Ph-	His
17-004	ALL-2	Ph-	His
17-020	Ph+3	Ph+	Asian
16-022	Ph+4	Ph+	Asian
UCI-001	Ph-like-5	Ph-like P2	His
UCI-002	Ph-like-6	Ph-like P2	His
UCI-004	Ph-like-9	Ph-like Ig	Asian
UCI-005	Ph-like-10	Ph-like Ig	Non-His
UCI-006	Ph-like-3	Ph-like Ig	His
NP-001	Ph-like-7	Ph-like P2	His
NP-004	Ph-like-4	Ph-like Ig	His
NP-005	Ph-like-8	Ph-like P2	His
p-001	H-W-4	N/A	White
p-002	H-H/L-1	N/A	His
p-003	H-H/L-2	N/A	His
p-004	H-H/L-3	N/A	His
p-005	H-W-1	N/A	White
p-006	H-A-1	N/A	Asian
p-007	H-A-2	N/A	Asian
p-008	H-H/L-4	N/A	His
p-009	H-W-2	N/A	White
p-010	H-W-3	N/A	White
p-011	H-H/L-5	N/A	His

The additional patient samples were from blood discards and were usable for preparing gDNA for our dPCR analysis. This allowed us to create an entirely new Figure 5 based on the expanded samples that show differences between *CRLF2::IGH* and *P2RY8::CRLF2* patients where loss of dPCR probe binding to AID target sequences is significantly higher in the *CRLF2::IGH* cases. We were also able to expand the analysis for *CRLF2* expression and added AID (*AICDA*) expression as well.

Figure 5: Rangel, et al

Fig. 4A and 4B

General remarks:

- the authors should use modern gene terminology (e.g., in Table 1), and the double colon notation for fusion genes: *BCR::ABL1* instead of *BCR-ABL*, *KMT2A::AFF1* instead of *MLL-AF4*, etc.

Thank you for pointing this out. We have changed the nomenclature in Table 1 and elsewhere in the manuscript to be more modern.

- Some minor typos: line 389 "associate with ALL", line 400 "preformed"

Thank you, we have corrected these and thoroughly proofread the revised manuscript.

Reviewer #2 (Remarks to the Author):

In this manuscript, Rangel et al have developed a digital PCR (dPCR) assay to track AID-induced lesions at two loci-CRLF2 and Bcl2-that are known to be targeted by AID in the context of B cell ALL. Using this technique, the authors were able to detect AID signatures in ALL patients. The authors posit that dPCR can be a convenient tool to detect AID-induced mutations in ALL patients and could contribute to the early detection of the disease, which is disproportionately affects the Latin American population.

AID-dependent lesions have been strongly associated with mutations and translocations associated with many types of B cell malignancies and the idea that (inadvertent or physiological) AID expression in developing B cells cooperate with RAG-induced lesions has been proposed for years; however, there is not enough data to support this idea. Thus, a robust method to detect mutations at the CRLF2 and Bcl2 regions is of much relevance. However, the authors should address the following issues to increase the impact of the work.

We thank Reviewer 2 for their comments and highlighting the relevance and importance of our work.

Fig. 1. The authors note that AID expression in the Nalm6 cells is below the level of detection, yet they find signatures of AID activity. It is unclear if these truly represent AID activity at these regions or the activity of some other APOBECs or just replication-induced lesions. The authors should attempt to delete AID from the Nalm6 or Reh lines they have used to examine if the mutations detected are truly AID-dependent.

We definitely understand the importance of showing this direct link with AID activity and indels at ABC sites. As the signatures detected by amplicon sequencing in Nalm6 cells have likely arisen over many years of culturing (we have detected very similar patterns from Nalm6 clones cultured for long or short periods of time), knocking out AID may not reduce or eliminate the number of reads with indels. To address this critique, we have extensively revised Figure 3 and added a time course study of AID induced by doxycycline as new Fig. 3E and 3F (below). This directly shows by dPCR that the longer AID is active in these cells, the more mutations occur at both the *CRLF2* and *BCL2* ABCs. We also performed amplicon sequencing after the time course. In new Fig. 3G, we show the original AmpSeq deletion pattern in blue and the pattern following 120 hours of AID activity overlaid in red that demonstrates a higher level of deletions in the same sequence with the CpGs in the *CRLF2* ABC. Thus, while we cannot fully rule out that other APOBECs or replication-induced lesions may have some contributions, we feel that these deletions accumulating within 5 days of AID overexpression strongly supports AID as the cause of damage.

Figure 3: Rangel, et al

Fig. 3. Here the authors try to correlate AID-induced mutations at CRLF2 and Bcl-2 in Nalm6 cells over-expressing AID from a lentiviral vector. This is one of the stronger figures of the manuscript. However, it is unclear how specific AID-induced mutations are for the two test regions, given that AID has been shown to mutate many sequences beyond the Ig loci. The authors used a couple of control regions but it is unclear how much power they have in this rather restricted analysis to conclude that CRLF2 and Bcl-2 are specific AID targets that can be detected by dPCR.

We focus on the *CRFL2* and *BCL2* ABCs as these are the regions where DNA DSBs form leading to *IGH* chromosomal translocations linked to cancer. We completely agree with the reviewer that AID is most likely having activity outside of these regions. Indeed, we feel that

signatures of high AID activity may occur genome-wide in patients with Ph-like ALL. Activity at oncogenes that lead to translocations is selected for due to the proliferative capacity conferred, but we expect that these cells would have evidence of this activity elsewhere in the genome. As we accumulate more patient samples with matched normal controls, future work will focus on WGS and looking for known COSMIC signatures of AID activity. The purpose of the *AMIGO1* and *PLEKHA5* analysis here was to show using a set of dPCR probes we had available that not all WCGW or CpG sites in the genome are attacked by AID. We realize this is an extremely limited sample and we do not want to suggest that no additional AID-induced lesions are occurring. Thus, we have removed this data as we feel it does not add to the story. We feel that it is also important to point out that the dPCR for *CRLF2* and *BCL2* already shows that there is specificity regarding which of the AID target sequences get mutated. At both loci, the Cy5 probe binds to a region of the amplicon with WCGW and CpG sites, but there is very little drop-off and the levels are comparable to the SUN probe that binds where there are no AID target sequences.

To further address Review 2's comment, we expanded the HTGTS analysis to include cells following AID expression. The idea being that this will not only map translocations that occur due to high AID expression, but also reveal sites in the genome vulnerable to AID-induced damage. We did see that dox leading to AID expression doubled the number of mutations mapped in the genome (compare control [Ctr_Rep3] to dox-induced [Dox_Rep3] below with filled circles showing number of variants). The most variants are mapped to Chr 14 as this is where the bait DSB was induced. AID expression did not lead to additional translocations, however. This is not completely surprising as our model requires both AID creating lesions and a nuclease like Artemis to cut the lesion to make a DSB (PMID: 35094960, 29735400). This is something we are actively testing and will be the focus of future work using the HTGTS method, which is why we did not include this data in our current revision.

Ctr_Rep3

Displaying 2,077 of 2,089 hits
hg38 - 2.5 Mb bins

Dox_Rep3

Displaying 3,942 of 3,967 hits
hg38 - 2.5 Mb bins

reviewers only

In summary, we have attempted to clarify that while AID activity at *CRLF2* and *BCL2* is our focus due to their oncogenic potential, AID is likely targeting other regions, the effect of which is unknown regarding the etiology to Ph-like ALL and is the focus of ongoing study in the lab.

Fig 4 and 5 employ dPCR to examine the mutation profile of the target genes. The authors clearly can use the technique to pick up mutations, but again the results are only correlative as there is no data on AID expression. Unless there is a way to detect AID expression, even by qPCR, it is difficult to interpret the data.

We agree with the reviewer that demonstrating high AID expression in the patients would be strong evidence of this link. For the expanded set of the patients where we had RNA available, we performed qPCR and added this data as new Fig. 4B.

Fig. 4A

Fig. 4B

One reason we did not perform this analysis in the 1st submission is that it is difficult to ascertain what it means to detect high levels of AID expression at one point in time from a patient sample. The *CRLF2::IGH* translocation occurs in pre-B cell development and AID expression may be transiently high at this time to allow for translocation formation but may not remain high. This is why we chose to put greater emphasis on the mutation signatures left by AID activity in new Figure 5 using dPCR at the *CRLF2* and *BCL2* loci as mutations and INDELS at these CpG sites act as information scars from past AID expression. Still the results are very telling as Ph-like-1 has very high *AICDA* and *CRLF2* expression as does Ph+2. Ph+2 was the focus of our response to Reviewer #1's comment 6 as it had an abnormal dPCR and *CRLF2* expression profile that made it seem more Ph-like, but we could not confirm a *CRLF2* rearrangement. AID expression is lower for the Ph+ and ALL patients, as expected, but higher in Ph-like-6 and elevated in Ph-like-2 and -4 as well.

Fig. 6 compares the transcriptome of Ph-ALL patient cohorts. It is unclear what the data suggests. There are bound to be transcriptomic differences between different patients. And again, in the absence of any data suggesting AID is expressed in these leukemias, it is difficult to interpret the data.

Our main goal in Figures 4D and 4E was to highlight the differences between the Ph-like and Ph+ patients. It is true that different patients will have very different expression profiles, but we feel it is important to highlight the DEGs among patients from these two populations. It may also benefit other studies looking to identify markers of Ph-like ALL. As this is a relatively rare patient population, making as much of the information publicly available is critical to the field. We do agree that not all of the data may be informative. We have moved this information to an earlier part of the manuscript where we provide information on our patient cohort. We have also chosen to keep only the GO Biological Process terms in the main text and moved the Reactome and DSigDB data to the supplement. The relevance of the GO Biological Process terms for Ph-

like ALL is due to recent studies showing that mullite kinase pathways may be active in these patients leading to drug resistance on only targeting a single pathway (PMID: 32191635, 38657263).

D Pathway Enrichment of DEGs from Ph-Like Cohort

E Pathway Enrichment of DEGs from Ph+ Cohort

Fig. 4D and 4C

Reviewer #3 (Remarks to the Author):

The manuscript by Ranger and Sterrenberg et al establishes a new digital PCR (dPCR) assay to detect the mutational landscape characterising acute lymphoblastic leukaemia (ALL). The authors focus their study on a subtype of ALL that shows increased prevalence in Latin Americans and lacks BCR-ABL1 chromosomal translocations typical of Philadelphia-positive ALL; yet shares similar patterns of gene expression with Ph positive ALL, therefore referred to as Ph-like ALL. An established dPCR-based assay would constitute a rapid and easy-to-access diagnostics method for Ph-like ALL, possibly improving early detection and treatment of this heterogeneous group of ALL patients for which standard treatment is often a poor choice. Similarly, a dPCR carries the potential to shed light into the aetiology of Ph-like ALL. Evidence suggests a role for AID-induced mutation in bone marrow B-cell precursors, and dPCR allows a more detailed analysis of the regions where AID-induced double-strand DNA breaks are known to occur.

The authors initially use cell lines to characterise regions of high AID-induced mutational burden - AID break clusters (ABCs) - in CRLF2 and BCL2 loci, comparing amplicon sequencing to dPCR. Overexpression of AID is used to determine whether AID activity correlates with

increased genomic instability at CpG sites on *CRLF2* and *BCL2* ABCs. A few questions arise from these analyses:

1. Do the levels of control probe SUN stay constant with AID overexpression? I note that dPCR in *CRLF2* ABC shows that TAMRA and Cy5 probes have increased drop-off with AID overexpression and SUN probe is located between the two. Depending on the DNA rearrangements at play, there may be a decrease in the signal for SUN.

The reviewer brings up a very important point that we have carefully considered as we develop the dPCR assay. The major reason that we included the SUN-labeled probe for *CRLF2* (and HEX-labeled probe for *BCL2*) was to determine if a region with no AID target sequence changed. Based on the reviewer's comment, we felt it was important to explicitly show this and have added the data for SUN and HEX to the new Figure 3. For the dose response data in Fig. 3C and 3D, the values for SUN/HEX drop-off change very little and the FAM and TAMRA (*CRLF2*) and FAM and Cy5 (*BCL2*) drop-off levels are significantly higher. Similar results are measured in the time-course. SUN/HEX values increase in the later time points but remain significantly lower than the other probe signals, except for Cy5, indicating little AID targeting of these regions compared to the other preferred sites in the region.

Fig. 3C-3F

Speaking directly to the reviewer's comment, the dPCR of the patient data varied, but for Ph-like-4 and Ph+2, we measure very high SUN drop-off that we feel is indicative of rearrangements at *CRLF2* (Fig. 5B). As we indicate in the text, we unfortunately did not have enough material from Ph+2 to fully confirm this:

For the Ph+2 sample, we unfortunately did not have adequate DNA from this patient to reach the same read depth, but there are several notable features from this analysis. For one, the expected BCR::ABL1 translocation is clearly present (Fig. 5D). Overall, there are far fewer SVs in Ph+2 compared to Ph-like-4 (Supplementary Table

4). Also, instability on chromosome 14 in Ph+2 is consistent with RAG activity resulting in rearrangements at the IGH locus and may be a distinguishing feature between Ph+ALL and Ph+ myeloid leukemias⁶². Interestingly, we do not see instability at the CRLF2 locus on the X chromosome even though dPCR (Fig. 5B) indicates a deletion or mutation at this site and qPCR shows high CRLF2 (Fig. 4A) and AICDA (Fig. 4B) expression. Based on this evidence, we cannot completely rule out that a CRLF2 rearrangement is present, but not detected due to low read coverage and depth. A complete listing of annotated variants is shown in Supplementary Table 5 for Ph-like-4 and Supplementary Table 6 for Ph+2.

Fig. 5B

2. dPCR is based on fluorescent signal in individual drops/DNA molecules. How do the various fluorescent probe signals compare to each other? Do the authors find evidence for particular combinations of probe drop-offs, i.e. do they happen on the same molecule? There is a greater level of detail here that is not explored/showed and could be informative.

This is a very intriguing possibility, and we agree with the reviewer that this may allow for a greater level of clarity, especially regarding the drop-off assays. The instrument we use is the Absolute-Q from Thermo Fisher Scientific. It differs from the Bio-Rad droplet digital PCR machine in that instead of molecules being encased in individual droplets, they are partitioned into over 20,000 separated microwells on a microfluidic array plate (MAP) (see part of Fig. 2A below). Amplification and acquisition of fluorescent signal is done simultaneously with each of the channels measuring signal from each well. We reached out to the field application specialist at Thermo Fisher to see if it is possible to overlay the signals to see which signals are coming from which well to determine changes in probe binding. The response was that, "the instrument is capable of displaying relative fluorescence for each individual microchamber but it doesn't tell you which microchamber is corresponding to what relative fluorescence." This may only be a software limitation that can be overcome in the future. Still, even with this upgrade, this may not give molecule level resolution as we cannot rule out multiple molecules in a single well. This same limitation applies to droplet dPCR. We are currently working on this for future assay designs where we will attempt to use limiting amounts of DNA to achieve close to single molecule resolution.

Quantify signal

Fig. 2A

3. Mapping of AmpSeq primers for the BCL2 ABC (Supplementary Figure 1B) would be helpful; the point about the SUN probe made earlier also applies here, particularly with amplicon sequencing showing rearrangement downstream of the 3rd CpG site/FAM probe.

We thank the reviewer for this suggestion. We have added the AmpSeq primers for *BCL2* to Supplementary Figure 1B (AmpSeq in orange and dPCR in yellow). As explained above the HEX signal downstream of the 3rd CpG changes very little with AID expression in dPCR. The AmpSeq data in its entirety still show readable sequence details. So, we have also included here for the reviewer a blown-up version of the raw data showing that there are not indels detected in untreated Nalm6 cells where the HEX probe is binding downstream of the FAM probe.

```

CCTTTAGAGAGTTGCTTTACGTGGCCTGTTTCAACACAGACCACCAGAGCCCTCCT -58
GCCCTCCTTCCGCGGGGGCTTTCTCATGGCTGTCCTTCAGGGTCTTCTGAAATGCAG -116
TGGTGCTTACGCTCCACCAAGAAAGCAGGAAACCTGTGGTATGAAGCCAGACCTCCCC -174
      Cy5                                FAM
GGCGGCGCTCAGGGAACAGAATGATCAGACCTTTGAATGATTCTAATTTTAAAGCAAA -232
ATATTATTTTATGAAAGGTTTACATTGTCAAAGTGATGAATATGGAATATCCAATCCT -290
      HEX
GTGCTGCTATCCTGCCAAAATCATTTTAATGGAGTCAGTTTGCAGTATGCTCCACGTG -348
GTAAGATCCTCCAAGCTGCTTTAGAAGTAA -378
    
```

Fig. S1B

4. It's unclear how the LAM-HTGTS results constitute a significant addition to the manuscript ; similar analyses in Nalm6 cells with AID overexpression could inform whether increased genomic instability brought about by AID-induced mutation results in chromosomal translocations.

We agree with this critique from the reviewer. The major use for HTGTS in our 1st submission is to verify that our sgRNAs for Cas9 are cutting efficiently and that use of both sgRNAs at *IGH* and *CRLF2* or *BCL2* can lead to a translocation. Also, it shows that Nalm6-Cas9 cells are not prone to extensive off target cutting. As this is a verification of the dPCR, we felt it more appropriate for the supplement and have moved it to Supplementary Figure 3. We completely agree that HTGTS with AID overexpression would be an important experiment and in response to reviewer comments, we performed this analysis. This was also a critique from Reviewer 2 and we address it above, but restate our results here for the benefit of Reviewer 3:

We expanded the HTGTS analysis to include cells following AID expression. The idea being that this will not only map translocations that occur due to high AID expression, but also reveal sites in the genome vulnerable to AID-induced damage. We did see that dox leading to AID expression doubled the number of mutations mapped in the genome (compare control [Ctr_Rep3] to dox-induced [Dox_Rep3] below with filled circles showing number of variants). The most variants are mapped to Chr 14 as this is where the bait DSB was induced. AID expression did not lead to additional translocations, however. This is not surprising as our model requires both AID creating lesions and a nuclease like Artemis to cut the lesion to make a DSB (PMID: 35094960, 29735400). This is something we are actively testing and will be the focus of future work using the HTGTS method, which is why we did not include this data in our current revision.

In summary, we have attempted to clarify that while AID activity at *CRLF2* and *BCL2* is our focus due to their oncogenic potential, AID is likely targeting other regions, the effect of which is unknown regarding the etiology to Ph-like ALL and is the focus of ongoing study in the lab.

Ctr_Rep3

Displaying 2,077 of 2,089 hits
hg38 - 2.5 Mb bins

Dox_Rep3

Displaying 3,942 of 3,967 hits
hg38 - 2.5 Mb bins

reviewers only

When analysing patient samples, the authors provide a compelling case for the outperformance of dPCR over amplicon sequencing, but only two Ph-like ALL samples are used.

This was a significant critique from all reviewers. In response, we have expanded the number of Ph-like ALL patients in our study from 2 to 10, which includes 4 Hispanics with *CRLF2* and *IGH* rearrangements and 4 Hispanics with *P2RY8* and *CRLF2* rearrangements. As this increase in the number of samples made use of the coded IDs more difficult, we have attempted to simplify the sample names. We provide a key here for the reviewer to be able to refer back to the names used in the 1st submission.

Table 1. De-Identified Material Collected from ALL Patients Treated at the University of California, Irvine

Sample	Age	Sex	Ethnicity	Genetics/FISH Findings	Source	% Blasts
Ph-Like ALL¹						
Ph-like-1	60	M	Hispanic	CRLF2 and IGH Rearrangement ²	BM	95
Ph-like-2	24	M	Hispanic	CRLF2 and IGH Rearrangement ²	BM	95
Ph-like-3	70	F	Hispanic	CRLF2 and IGH Rearrangement ²	BM	95
Ph-like-4	54	M	Hispanic	CRLF2::IGH ³	PB	90
Ph-like-5	31	F	Hispanic	P2RY8 and CRLF2 Rearrangement ²	BM	62
Ph-like-6	46	F	Hispanic	P2RY8 and CRLF2 Rearrangement ²	BM	50
Ph-like-7	70	M	Hispanic	P2RY8::CRLF2 ⁴	PB	59
Ph-like-8	26	M	Hispanic	P2RY8 and CRLF2 Rearrangement ²	PB	90
Ph-like-9	29	M	Asian	CRLF2 and IGH Rearrangement ²	BM	56
Ph-like-10	75	M	White	CRLF2 and IGH Rearrangement ²	PB	91
Ph-Positive ALL						
Ph+1	64	M	Hispanic	BCR::ABL1	BM	95
Ph+2	44	M	Hispanic	BCR::ABL1	BM	90
Ph+3	78	F	Asian	BCR::ABL1	BM	90
Ph+4	50	F	Asian	BCR::ABL1	BM	90
Ph-Negative ALL						
ALL-1	48	M	Hispanic	Negative for BCR::ABL1 , IGH , CRLF2 , MYC , and KMT2A Rearrangements	PB	73
ALL-2	61	M	Hispanic	KMT2A::AFF1	BM	80

¹Ph-like ALL was diagnosed using flow cytometry to detect increased *CRLF2* expression.

²Break-apart probes were used to detect *CRLF2* and *IGH* rearrangements

³Confirmed by long-read sequencing in this work (see Fig. 5C)

⁴Next generation sequencing was used to detect a *P2RY8::CRLF2* fusion RNA transcript

Patient Key (reviewers only)

Sample	NEW	Ph	Race/Eth
17-062	Ph-like-1	Ph-like Ig	His
19-021	Ph-like-2	Ph-like Ig	His
15-010	Ph+1	Ph+	His
17-061	Ph+2	Ph+	His
19-035	ALL-1	Ph-	His
17-004	ALL-2	Ph-	His
17-020	Ph+3	Ph+	Asian
16-022	Ph+4	Ph+	Asian
UCI-001	Ph-like-5	Ph-like P2	His
UCI-002	Ph-like-6	Ph-like P2	His
UCI-004	Ph-like-9	Ph-like Ig	Asian
UCI-005	Ph-like-10	Ph-like Ig	Non-His
UCI-006	Ph-like-3	Ph-like Ig	His
NP-001	Ph-like-7	Ph-like P2	His
NP-004	Ph-like-4	Ph-like Ig	His
NP-005	Ph-like-8	Ph-like P2	His
p-001	H-W-4	N/A	White
p-002	H-H/L-1	N/A	His
p-003	H-H/L-2	N/A	His
p-004	H-H/L-3	N/A	His
p-005	H-W-1	N/A	White
p-006	H-A-1	N/A	Asian
p-007	H-A-2	N/A	Asian
p-008	H-H/L-4	N/A	His
p-009	H-W-2	N/A	White
p-010	H-W-3	N/A	White
p-011	H-H/L-5	N/A	His

1. It is unclear how predictive value of dPCR analyses in the *CRLF2* ABC regions for Ph-like ALL - i.e. drop-off rates can vary within patients of the same group, and yet show similar patterns between patients of different groups (for eg, patients 17-062 (Ph-like) and 15-010 (Ph positive) show very similar drop-off levels in Figures 4D and E). While it is understandable that Ph-like ALL samples are rare the small-scale study carried out by the authors doesn't provide evidence that dPCR can discriminate between Ph-like and Ph positive ALL samples.

We agree with the reviewer that the samples we were comparing in the 1st submission did not show substantial differences in drop-off between the Ph- and Ph+ samples, only with the cancer versus healthy samples. However, with the greater number of patients we have included here, we can make much more significant comparisons. Specifically, we can now compare the *CRLF2::IGH* Ph-like ALL patients to the *P2RY8::CRLF2* patients. Previous studies (PMID: 20847213) have shown that the *CRLF2-IGH* translocation is likely AID driven while the

P2RY8::CRLF2 deletion is not and here we clearly see increased drop off at the AID target regions of *CRLF2* in the patients with the translocation, but not the intrachromosomal deletion.

Fig. 5A

2. Although figures 4 and 5 refer to patients 17-062 and 19-021 as Ph negative, I assume these are Ph-like as stated in Table 1. Do Ph negative patients (eg 17-004, which doesn't show increased expression of *CRLF2*) also show genomic instability at the *CRLF2* locus?

The reviewer is correct that 17-062 and 19-021 are Ph-like. As stated above, we have revised Table 1 to make the naming of patient samples easier for the reader, especially with the increased number of samples. We have utilized this in new Figures 4 and 5 to remove any ambiguity of Ph status. While we would have ideally also run dPCR on ALL-1 and ALL-2 (formally 17-004 and 19-035) we were unable to as these sample came from a biorepository that all researchers at UCI have access to and the only remaining samples from these patients with stored in Trizol for RNA extraction. Moving forward, we have our own IRB to collect and consent patients with and we have amended this to include any ALL patient regardless of Ph status so we can increase the number of Ph- samples in our studies.

3. The authors state in the manuscript 'dPCR could provide an early indicator of disease' but the analyses shown are performed in samples with an overload of B-ALL cells. There is no indication dPCR will have the sensitivity to detect small numbers of pre-ALL cells in individuals at risk of developing disease. This is an important consideration if dPCR were to be adopted for early detection.

We completely agree with the reviewer on this point. Our main goal here was to establish a dPCR system to detect differences at specific loci in different patients. We are currently developing a longitudinal study for patients diagnosed with Ph-like ALL where we can obtain samples during their chemotherapy treatment and into remission to monitor the status of *CRLF2* in their B cells. As part of this, we are also developing a dPCR probe set that is specific for the translocation itself so that instead of looking at drop off, we are monitoring for a specific rearrangement, which should greatly improve the sensitivity of the assay.

Overall, I find the study useful in showing dPCR is a powerful methodology to characterise genomic instability in ALL samples. But it doesn't provide compelling evidence supporting the

use of dPCR to discriminate between different ALL subtypes or detect pre-disease states and these are key requirements for dPCR to be adopted as a new diagnostic tool. Likewise, other than correlating AID overexpression with increased genomic instability at *CRLF2* and *BCL2* loci, this study doesn't significantly advance our understanding of the aetiology of ALL.

We thank the reviewer for their critical assessment of our work in pointing out the potential of the dPCR system, but also describing areas where the technique needs to improve. We feel that adding a much larger sample of ALL patients and healthy donors to our study allowed us to make more meaningful comparisons that demonstrate the potential for dPCR. In addition to Fig. 5A above comparing between the patients with different *CRLF2* rearrangements, we were also struck by the results of **Fig. 5E and 5F** that show *CRLF2* to be vulnerable to mutations in CD19+ cells compared to CD19- cells, but this was not the case at *BCL2*. While this was only significant in the healthy Hispanic population, there is also elevated drop-off in the non-Hispanic population, indicating that this region may be a common site of mutations in developing B cells. In terms of what this may mean for Ph-like ALL etiology in Hispanics will require further mechanistic studies. For example, we have preliminary data that the Hispanic population may carry mutations in the Artemis nuclease required to open hairpins created by the RAG nuclease at *IGH*. Slowing the repair kinetics of the *IGH* DSB, may potentiate translocations at *CRLF2* damaged by AID. Also of interest is that the ABC at *BCL2* does not seem to be vulnerable to these mutations during B cell development. What these results and other recent studies suggest is that factors beyond just DNA sequence are dictating sensitivity to AID and most likely DNA methylation is a factor (PMID: 25624348, 34260910). AID can deaminate both methylated and unmethylated cytosines. While AID converts C→U, it converts meC→T with these T's being repaired more slowly leading to a longer-lived lesion (PMID: 29735400, 38490833). While we have preliminary data supporting this, the role of DNA methylation in Ph-like ALL disparities requires more extensive studies and will be a focus of our future work.

REVIEWERS' COMMENTS

Reviewer #1 (Remarks to the Author):

1. The reviewer is unable to find a more detailed immunophenotype in Table 1 or the manuscript. The diagnosis of ALL was probably not simply based on the detection of CRLF2 expression or the genetic detection of a CRLF2::IGH fusion. What about other antigens like CD10, cyIg, CD34, CD33, TdT, etc? What were the immunophenotypes (pro, pre, common ALL)?

2. The authors have investigated five healthy Hispanic and six healthy non-Hispanic individuals (or patients? – if they are healthy, they should not be called "patients") by dPCR at the CRLF2 and BCL2 loci and found a higher level of mutations at CRLF2 in the CD19+ cell fraction. Realistically speaking this is still a relatively small number. The statistical power is much too small to prove anything. These observations can at best only provide a hint and this should be clearly stated in the manuscript. From these relatively small numbers it is impossible to derive conclusions about the Hispanic population in general. Therefore, the reviewer suggests modifying the manuscript title a little bit: "Increased AID Results in Mutations at the CRLF2 Locus may be Implicated in Latin American ALL Health Disparities", or similar.

The authors have basically answered the reviewer's questions adequately. The reviewer still thinks that the main critical point is the relatively low number of investigated individuals. If the authors want to suggest that a generally increased AID activity in Hispanics is the cause for the higher incidence of Ph-like ALL in this population, much larger numbers of individuals (Hispanics and non-Hispanics) would have to be investigated for AID activity. The general message of the manuscript should thus be formulated somewhat more cautiously.

Reviewer #2 (Remarks to the Author):

Authors have addressed most of the comments raised by this reviewer. The manuscript is now ready for publication.

Reviewer #3 (Remarks to the Author):

The authors provide a much improved manuscript addressing most of on the reviewers comments. The manuscript is clear in providing compelling evidence that the digital PCR is a more robust approach to characterize B-ALL samples with potential applications on diagnostics or to study the aetiology of the disease in longitudinal studies. One major weakness of the original version was the reduced number of patients and the authors have significantly improved this by including an additional 8 Ph-like ALL patients. The authors have included a considerable amount of new data and made their analyses clearer on the manuscript.

Point-by-Point Response to Reviewer Comments.

Rangel and Sterrenberg, et. al, *Increased AID Results in Mutations at the CRLF2 Locus Implicated in Latin American ALL Health Disparities*

We thank the reviewers for evaluating our revised manuscript. As both Reviewers 2 and 3 had no additional requests/critiques, we focus on the comments from Reviewer 1. Below is our detailed point-by-point response with original reviewer comments in **black** and our response in **blue**. We have also added new data where appropriate to facilitate evaluation of our responses.

Reviewer #1 (Remarks to the Author):

1. The reviewer is unable to find a more detailed immunophenotype in Table 1 or the manuscript. The diagnosis of ALL was probably not simply based on the detection of CRLF2 expression or the genetic detection of a CRLF2::IGH fusion. What about other antigens like CD10, cylg, CD34, CD33, TdT, etc? What were the immunophenotypes (pro, pre, common ALL)?

We apologize for not adding these details into the previous revision as we did not realize the extent of detail the reviewer was looking for. We have now added the full immunophenotype for all patients into Table 1 (see below).

Table 1. De-Identified Material Collected from ALL Patients Treated at the University of California, Irvine

Sample	Age Range	Sex	Ethnicity	Genetics/FISH Findings Immunophenotype	Source	% Blasts
Ph-Like ALL¹						
Ph-like-1	50-80	M	Hispanic	CRLF2 and IGH Rearrangement ² CD19+, CD19+, CD20+, CD34 (subset), CD38+, cCD22+, cCD79a, and TdT + CD19+	BM	95
Ph-like-2	20-50	M	Hispanic	CRLF2 and IGH Rearrangement ² CD19+, CD10+, CD20+, cCD79a+, TdT +, CD34 (subset), and HLA-DR (subset)	BM	95
Ph-like-3	50-80	F	Hispanic	CRLF2 and IGH Rearrangement ² CD34+, TdT +, CD10+, CD19+, CD22(subset), CD33+, cCD79a+, cCRLF2+, HLA-DR+, CD38(subset)	BM	95
Ph-like-4	50-80	M	Hispanic	CRLF2::IGH ² CD19+, CD10+, CD22 (SUBSET), CD34+, HLA-DR+, CD33 (SUBSET), CD20-	PB	90
Ph-like-5	20-50	F	Hispanic	P2RY8 and CRLF2 Rearrangement ² CD19+ CD34+ CD10(subset), CD22(very dim), CD38+ HLA-DR+	BM	62
Ph-like-6	20-50	F	Hispanic	P2RY8 and CRLF2 Rearrangement ² CD19+ CD34+ CD10(very small subset), CD22(subset), CD38(dim), CD11b+, cCD79a(dim), TdT (dim), cCRLF2(dim), HLA-DR+	BM	50
Ph-like-7	50-80	M	Hispanic	P2RY8::CRLF2 ⁴ CD19+, CD10+, CD20+, CD22+, TdT (subset), CD38+, CD34 (large subset), CD79a+, HLA-DR+	PB	59
Ph-like-8	20-50	M	Hispanic	P2RY8 and CRLF2 Rearrangement ² CD10+, CD19+, CD20+, CD22+, cCRLF2+, cCD79a+, CD34+, CD38+, HLA-DR+, and TdT +	PB	90
Ph-like-9	20-50	M	Asian	CRLF2 and IGH Rearrangement ² CD19+ CD34+ CD10+, CD22+, CD38+ HLA-DR, aberrant expression of CD33(subset)	BM	56
Ph-like-10	50-80	M	White	CRLF2 and IGH Rearrangement ² CD34+ TdT (large subset), CD19+, cCD79a+, CD10+, CD10+, CD20(subset, 45%), sCD22(subset), cCRLF2+, CD38+, HLA-DR+	PB	91
Ph-Positive ALL						
Ph+1	50-80	M	Hispanic	BCR::ABL1 CD19+, CD10+, CD34+, CD79a (cytoplasmic) and TdT (cytoplasmic) with expression of a myeloid marker of CD13	BM	95
Ph+2	20-50	M	Hispanic	BCR::ABL1 CD19+, CD10+, CD34+, HLA-DR+, CD33+, CD38+, cCD79a+, and TdT +	BM	90
Ph+3	50-80	F	Asian	BCR::ABL1 CD34+, CD10+, CD19+, CD79a+ and TdT + CD22(subset)	BM	90
Ph+4	50-80	F	Asian	BCR::ABL1 CD19+, CD10+, CD22 (subset), CD34+, HLA-DR+, CD33 (subset), CD20-	BM	90
Ph-Negative ALL						
ALL-1	20-50	M	Hispanic	Negative for BCR::ABL1 , IGH , CRLF2 , MYC , and KMT2A Rearrangements CD10+, CD20+, CD79a+, TdT +	PB	73
ALL-2	50-80	M	Hispanic	KMT2A::AFF1 partial CD33+, CD34+, CD38+, CD79a+, HLA-DR+ TdT +	BM	80

2. The authors have investigated five healthy Hispanic and six healthy non-Hispanic individuals (or patients? – if they are healthy, they should not be called "patients") by dPCR at the *CRLF2* and *BCL2* loci and found a higher level of mutations at *CRLF2* in the CD19+ cell fraction. Realistically speaking this is still a relatively small number. The statistical power is much too small to prove anything. These observations can at best only provide a hint and this should be clearly stated in the manuscript. From these relatively small numbers it is impossible to derive conclusions about the Hispanic population in general. Therefore, the reviewer suggests modifying the manuscript title a little bit: "Increased AID Results in Mutations at the *CRLF2* Locus may be Implicated in Latin American ALL Health Disparities", or similar.

The authors have basically answered the reviewer's questions adequately. The reviewer still thinks that the main critical point is the relatively low number of investigated individuals. If the authors want to suggest that a generally increased AID activity in Hispanics is the cause for the higher incidence of Ph-like ALL in this population, much larger numbers of individuals (Hispanics and non-Hispanics) would have to be investigated for AID activity. The general message of the manuscript should thus be formulated somewhat more cautiously.

We have taken care in the manuscript to refer to the disease-free samples as “healthy donors” and not “patients”. Thank you for pointing this out. While we agree with the reviewer that the number of patient samples we have been able to collect is low, and therefore limits extrapolation to the Hispanic/Latino population in general, we still feel there are valuable insights, especially in establishing a dPCR method to look at mutation scars in patient samples. While the reviewer feels that the title is too provocative, we feel that it is already well-established in the literature that the *CRLF2* locus is implicated in Latin American health disparities and that our cell-based model clearly shows that increased AID significantly increases mutations at *CRLF2*, thus justifying it. We have added the following to the discussion to address the reviewer's concern, “It is important to note that confirmation of these results will need a larger pool of Hispanic and Latino Ph-like ALL patients beyond the limited number we were able to collect here and is a central goal of our future work.”

Reviewer #2 (Remarks to the Author):

Authors have addressed most of the comments raised by this reviewer. The manuscript is now ready for publication.

We thank Reviewer 2 for their insightful comments that have strengthened our work.

Reviewer #3 (Remarks to the Author):

The authors provide a much improved manuscript addressing most of on the reviewers comments. The manuscript is clear in providing compelling evidence that the digital PCR is a more robust approach to characterize B-ALL samples with potential applications on diagnostics or to study the aetiology of the disease in longitudinal studies. One major weakness of the original version was the reduced number of patients and the authors have significantly improved this by including an additional 8 Ph-like ALL patients. The authors have included a considerable amount of new data and made their analyses clearer on the manuscript.

We thank Reviewer 3 for their comments and suggestions. We hope to further build on this foundational work in the future.